# $NO_x$ emissions in France in 2019-2021 as estimated by the high spatial resolution assimilation of TROPOMI $NO_2$ observations

Robin Plauchu [1], Audrey Fortems-Cheiney [1,*], Grégoire Broquet [1], Isabelle Pison [1], Antoine Berchet [1], Elise Potier [1,*], Gaëlle Dufour [2], Adriana Coman [3], Dilek Savas [2], Guillaume Siour [3], and Henk Eskes [4]

[1]Laboratoire des Sciences du Climat et de l'Environnement, LSCE/IPSL, CEA-CNRS-UVSQ, Université Paris-Saclay, F-91191 Gif-sur-Yvette, France
[2]Université Paris Cité and Univ Paris Est Créteil, CNRS, LISA, F-75013 Paris, France
[3]Univ Paris Est Créteil and Université Paris Cité, CNRS, LISA, F-94010 Créteil, France
[4]Royal Netherlands Meteorological Institute (KNMI), De Bilt, the Netherlands
[*]Now in Science Partners, Quai de Jemmapes, 75010 Paris, France

Correspondence to: Robin Plauchu (robin.plauchu@lsce.ipsl.fr)

**Abstract.**

Since 2018, TROPOMI on-board Sentinel-5P provides unprecedented images of $NO_2$ tropospheric columns at a relatively high spatial resolution with a daily revisit. This study aims at assessing the potential of the TROPOMI-PAL data to estimate the

national to urban $NO_x$ emissions in France from 2019 to 2021, using the variational mode of the recent Community Inversion Framework coupled to the CHIMERE regional transport model at a spatial resolution of $10\times10$ $km^2$. The seasonal to inter-annual variations of the $NO_x$ French emissions are analyzed. A specific attention is paid to the current capability to quantify strong anomalies in the $NO_x$ emissions at intra-annual scales such as the ones due to the COVID-19 pandemic, by using TROPOMI $NO_2$ observations.

At the annual scale, the inversions suggest a decrease of the average emissions over 2019-2021 of -3 % compared to the national budget from the Copernicus Atmosphere Monitoring Service regional inventory (CAMS-REG) for the year 2016, which is used as a prior estimate of the national scale emissions for each year by the Bayesian inversion framework. This is lower than the decrease of -14 % from 2016 to the average over 2019-2021 in the estimates of the French Technical Center for Air Pollution and Climate Change (CITEPA). The lower decrease in the inversion results may be linked for a large part to the limited level

of constraint brought by the TROPOMI data, due to the observation coverage and the ratio between the current level of errors in the observation and the chemistry-transport model, and the $NO_2$ signal from the French anthropogenic sources.

Focusing on local analysis and selecting the days during which the TROPOMI coverage is good over a specific local source, we compute the reductions in the $NO_x$ anthropogenic emission estimates by the inversions from spring 2019 to spring 2020. These reductions are particularly pronounced for the largest French urban areas with high emission levels (e.g., -26 % from

April 2019 to April 2020 in the Paris urban area), reflecting reductions in the intensity of vehicle traffic reported during the lockdown period. However, the system does not show large emission decreases for some of the largest cities in France (such as Bordeaux, Nice and Toulouse), even though they were also impacted by the lockdown measures.

Despite the current limitations for the monitoring of emissions at the national scale, or for some of the largest cities in France,

these results open positive perspectives regarding the ability to support the validation or improvement of inventories with satellite observations, at least at the local level. This leads to discussions on the need for a stepwise improvement of the inversion configuration for a better extraction and extrapolation in space and time of the information from the satellite observations.

# 1 Introduction

In Europe, nitrogen dioxide ($NO_2$) is emitted mainly by road traffic, thermal power plants and industrial activities and produced in the atmosphere by the oxidation of nitric oxide (NO), which is emitted by the same activities. $NO_2$ is of great interest due to its important role in many atmospheric processes with strong implications for air quality, health, climate change and ecosystems. It is one of the major air pollutants with adverse impact on health (Costa et al., 2014; EEA, 2020). Deposition of nitrogen compounds like nitrates, for which $NO_2$ is a precursor, leads to eutrophication of ecosystems (Stevens et al., 2018). $NO_2$ also indirectly affects the radiative forcing as a precursor of tropospheric ozone and particulate matter. $NO_2$ is therefore one of the regulated air quality pollutants. Nevertheless, despite ongoing improvements in the overall air quality, levels of air pollutants above standards of the European Union (EU) are still measured across Europe and air pollution remains a major health concern for European citizens (EEA, 2023). For example, France was condemned in 2019 by the Court of Justice of the EU (CJEU) for non-compliance with Directive 2008/50/EC relating to ambient air quality, and more specifically for exceeding systematically and persistently concentration limit values (CLV, 40 µg.m$^{-3}$ on annual average) for $NO_2$, particularly in the Ile-de-France area, close to traffic. According to Airparif (2022), the planned reductions in the emissions of nitrogen oxides ($NO_x$=NO+$NO_2$) will still be insufficient by 2025 to respect the $NO_2$ CLV, which should have been reached by January 2010. The society is thus faced with a major environmental challenge: the need to rapidly reduce $NO_2$ concentrations to levels that comply with the law (EU Directive 2016/2284) and do not impact human health or ecosystems and therefore to reduce anthropogenic $NO_x$ emissions. An accurate account of $NO_x$ emissions in space and time is needed to assess the effectiveness of policies aiming at reducing $NO_x$ emissions. However, the quantification of anthropogenic $NO_x$ emissions following a bottom-up (BU) approach, based on the statistics of activity sectors and fuel consumption and relying on emission factors per activity type, suffers from relatively large uncertainties. For example, at national and annual scales, these uncertainties reach 50-200 % depending on the activity sector in the European Monitoring and Evaluation Programme (EMEP) inventory (Kuenen and Dore, 2019) and these emission factors can be biased (e.g., with the Dieselgate, Brand (2016)). Schindlbacher et al. (2021) reported uncertainty estimates for national total $NO_x$ emissions ranging from 5 % for Norway to 45 % for Ireland. In addition, the use of proxies and typical temporal profiles inevitably introduces errors in the quantification of the spatio-temporal variability at high resolutions. In situ information could be used to analyze local emissions and their variations due to specific events or measures (Guevara et al., 2023). Finally, the BU inventories are often delivered with a 2-year lag. The assessment of AQ (Air Quality) policies (as mentioned above) would benefit from accurate emission inventories spatialized at a high resolution with a fast update capability and integrating independent information could play a critical role for the AQ analysis and policies. The use of atmospheric measurements to complement current BU approaches may then support the development of such inventories. Since the 2000s, $NO_2$ atmospheric mixing ratios have been monitored around the world by space-borne instruments, such as the Global Ozone Monitoring Experiment GOME (Burrows et al., 1999) and GOME-2 (Munro et al., 2016), the SCanning Imaging Absorption spectroMeter for Atmospheric CHartographY SCIAMACHY (Burrows et al., 1995; Bovensmann et al., 1999) and the Ozone Monitoring Instrument OMI (Levelt et al., 2018).

In this context, attempts have been made to develop so-called top-down (TD) methods, complementary to BU inventories, to

deduce NO$_x$ emissions from NO$_2$ satellite data. These methods are based on the principle that atmospheric levels and variations of NO$_2$ reflect the convolution of the amplitude and variations of NO$_x$ emissions with the atmospheric chemistry and physics. Through the statistical inverse modeling of the atmospheric chemistry and transport, one can derive estimates of the emissions based on the concentration fields. However, strong non-linear relationships exist between NO$_x$ emissions and satellite NO$_2$ tropospheric vertical column densities (TVCDs) (Lamsal et al., 2011; Vinken et al., 2014; Miyazaki et al., 2017; Elguindi et al., 2020) due to the complex chemistry affecting NO$_x$ in the atmosphere.

This complex atmospheric chemistry has been taken into account in various ways in more or less complex TD methods. Mass–balance approaches have been performed at the global (Lamsal et al., 2011; Vinken et al., 2014) and regional (Visser et al., 2019) scales, accounting for non-linear relationships between NO$_x$ emission changes and NO$_2$ TVCDs via reactions with hydroxyl radicals (OH) but with simple scaling factors. However, Stavrakou et al. (2013) have shown that other direct or indirect NO$_x$ sinks associated with other species (such as ozone O$_3$ or the HO$_2$ radical) could significantly influence NO$_2$ concentrations in the atmosphere and therefore TD estimates. A more detailed account of the complex NO$_x$ chemistry should thus support more accurate derivations of NO$_x$ emissions from NO$_2$ satellite data. Therefore, more elaborated approaches using chemistry transport models (CTM) with ensemble Kalman filter inverse modeling techniques or variational approaches, have been used to infer NO$_x$ emissions at the global (Müller and Stavrakou, 2005; Miyazaki et al., 2017) or at the regional scale (van der A et al., 2008; Mijling and van der A, 2012; Mijling et al., 2013; Lin, 2012; Ding et al., 2017; Savas et al., 2023). However, NO$_x$ inversions from satellite NO$_2$ observations have resolution-dependent biases: coarse-resolution models are known to bear negative biases in NO$_2$ over large sources (Valin et al., 2011). It would be therefore essential to operate at finer spatial resolutions. In addition, monitoring NO$_x$ emissions implies monitoring hotspots of emissions (large urban and industrial areas or strong point sources), which concentrate much of the global emissions (57 % of the global population lives in urban areas as of 2022, Ritchie and Roser (2018)). Being able to monitor individually emission hotspots is required to assess the emission reduction policies since the scales they target range from the regional scale (e.g., that of the EU) to the country scale or even to that of smaller territory units e.g., cities. This requires in turn a high spatial and temporal resolution mapping of NO$_2$ concentrations.

Since 2017, the TROPOspheric Monitoring Instrument (TROPOMI, Veefkind et al. (2012)) on-board the Copernicus Sentinel-5 Precursor (S5P) monitors atmospheric NO$_2$ with a high-resolution imaging (pixel size of about 5.6×3.6 km$^2$ since August 2019), which should support the quantification of anthropogenic emissions at national to local scales. With a swath as wide as approximately 2600 km on ground, the TROPOMI instrument also provides an unprecedented daily coverage. It theoretically covers any point of the Earth 1 to 2 times a day. The need for surface solar irradiance, the cloud cover and the quality filtering limit the number of pixels in numerous images locally (e.g., at high latitudes) but the possibility of having a follow-up week by week (even day by day) remains for large portions of the globe.

To fully exploit these TROPOMI satellite images, variational inversion systems seem particularly adapted since they allow for solving high-dimensional problems (Elbern et al., 2000; Quélo et al., 2005; Pison et al., 2009; Henze et al., 2009; Cao et al., 2022), typically addressing the emission fluxes at high spatial and temporal resolutions and assimilating a large number of data, such as provided in TROPOMI images. The non-linearities of the chemistry of NO$_x$ can be dealt with in a variational inversion

framework driving a regional CTM using a manageable chemistry scheme to simulate $NO_2$ concentrations and whose adjoint code is available.

In this context, this study assesses the potential of the TROPOMI observations to inform about $NO_x$ emissions in France from 2019 to 2021 at national to urban scales. The primary target of the inversions and analysis are the anthropogenic emissions, i.e., mainly those due to the combustion of fossil and biofuels. The soil emissions (including the impact of agricultural practices), which represent a much smaller share of the total national emissions and which are more diffuse are assumed to be more difficult to diagnose and are kept as a secondary target of the inversions.

We use the high-dimensional variational inversion drivers of the recent Community Inversion Framework (CIF, Berchet et al. (2021)). The CIF drives a configuration of the CHIMERE regional CTM (Menut et al., 2013; Mailler et al., 2017) covering France at $10\times10$ $km^2$ spatial resolution, including a chemistry module taking into account the complex $NO_x$ chemistry in gas-phase and its non-linearities, and of its adjoint (Fortems-Cheiney et al., 2021). This relatively fine spatial resolution makes it possible to focus on the French largest urban areas. The period 2019–2021 covers the phase of the COVID-19 crisis in spring 2020 during which $NO_2$ concentrations and $NO_x$ anthropogenic emissions are expected to have significantly dropped over Europe (Bauwens et al., 2020; Menut et al., 2020; Diamond and Wood, 2020; Ordóñez et al., 2020; Petetin et al., 2020; Barré et al., 2021; Gaubert et al., 2021; Deroubaix et al., 2021; Lee et al., 2021; Souri et al., 2021; Levelt et al., 2022; Guevara et al., 2021, 2022, 2023). In France, the population has been confined from March $17^{th}$ to May $10^{th}$ and all public spaces deemed non-essential to daily life in the country have been shut down. Then, from May $11^{th}$ to June $1^{st}$, lockdown restrictions have progressively been lifted. The population has been confined again from October $30^{th}$ to December $15^{th}$. The analysis of the emissions from 2019–2021 should thus provide insights on the current capability to quantify strong anomalies in the $NO_x$ emissions at intra-annual scales by using satellite $NO_2$ observations.

Our configuration of the CHIMERE CTM, the $NO_2$ TROPOMI satellite observations, and the variational inversion framework and set up are described in Section 2. Section 3 presents the results of our study, including comparisons between TROPOMI-PAL $NO_2$ TVCDs and their CHIMERE simulated equivalents and the analysis of the spatio-temporal variability of the French $NO_x$ emissions. Our conclusions are given in Section 4.

## 2 Data and method

### 2.1 Prior estimates of the emission maps

The principle of the inversion is to correct *a priori* emission maps later on referred as "prior emissions". In this study, the inversion controls NO and $NO_2$ emissions based on the Bayesian update of a prior estimate of these emissions. Therefore, there is a need for independent maps of the NOx emissions to derive this prior estimate. There is also a need for estimates of the anthropogenic emissions of 15 species (including non-methane volatile organic compounds NMVOCs, carbon monoxide CO, etc.) that are used in the chemistry scheme of the atmospheric chemistry transport model, even though they are not controlled by the inversions (these emissions remain fixed during the inversion process, see Section 2.5).

Here, the prior estimates of $NO_x$ anthropogenic emissions are based on a combination of the Copernicus Atmosphere Monitoring Service Regional (CAMS-REG) emission inventory (Kuenen et al., 2022) for the year 2016 and of the Inventaire National Spatialisé (INS, Ministère de la transition écologique et solidaire (2012)) for the year 2012. Hereafter, the term anthropogenic emissions mainly correspond to emissions due to the combustion of fossil and biofuels, and it excludes the emissions due to the soil fertilization by agriculture, such as the CAMS-REG and INS.

CAMS-REG is an inventory of the anthropogenic emissions of pollutants in Europe which is spatialized at a 0.05 ° longitude × 0.1 ° latitude resolution. Annual and national budgets in this inventory are based on the officially reported emission data by European countries to the Convention on Long-Range Transboundary Air Pollution and the EU National Emission Ceilings Directive. These budgets are then disaggregated based on proxies of the different sectors, separating point sources and areas sources, described in Kuenen et al. (2022). Default profiles for typical emission height by source type (Kuenen et al., 2022), which accounts for the average effective emission height (including plume rise), based on Bieser et al. (2011). Temporal disaggregation is based on temporal profiles provided per sector with typical month to month, weekday to week-end and diurnal variations (Ebel et al., 1997; Menut et al., 2012).

The prior estimates of the anthropogenic emissions are completely derived from the CAMS-REG inventory outside France. In France, it is derived from the annual budgets and the temporal and vertical profiles of the CAMS-REG inventory, but the horizontal spatialization is based on the INS, i.e. using proxies at municipal scale from this inventory. The inventory product used for the prior estimate of the emissions is therefore called "CAMS-REG/INS" in the following.

Following the GENEMIS recommendations (Kurtenbach et al., 2001; Aumont et al., 2003), the $NO_x$ anthropogenic emissions have been speciated as 90 % of NO, 9.2 % of $NO_2$, and 0.8 % of nitrous acid (HONO) emissions.

The NO biogenic soil emissions are prescribed using simulations from the Model of Emissions of Gases and Aerosols from Nature (MEGAN) model (Guenther et al., 2006), with a ~1×1 $km^2$ spatial resolution, which, in principle, does not take the impact of agricultural practices into account, even though it covers both natural and agricultural areas. Recent studies have indeed shown that MEGAN significantly underestimates soil emissions in agricultural areas (Oikawa et al., 2015; Almaraz et al., 2018; Sha et al., 2021; Zhu et al., 2023). However, there are large uncertainties in the $NO_x$ emissions due to agriculture, and in principle, there could be some overlapping between the agricultural and purely natural soil $NO_x$ emission estimates. It explains why these emissions are not provided by the CAMS-REG inventory (Kuenen et al., 2022). Therefore, we do not

include a specific agricultural soil NO$_x$ emissions component in our prior estimation of the NO$_x$ emissions.

Lightning NO$_x$ fluxes, whose impact on NO$_2$ concentrations is very small in Europe even in summer (Menut et al., 2020), are not accounted for. Fire emissions are also ignored, as we assume that they only slightly contribute to the NO$_x$ total emissions. These emission maps have been aggregated to the grid of CHIMERE for the years 2019 to 2021 (Section 2.2 and Figure 1).

## 2.2   Configuration of the CHIMERE CTM

We use the CHIMERE v2013 model to simulate fields of concentrations of gaseous chemical species in a domain that covers
France and its vicinity (11 °W-12 °E; 39.5 °N-54.5 °N, see Figure 1). The model horizontal grid is zoomed (Siour et al., 2013), with a 10 km resolution regular sub-grid in the center of the domain covering the full France and a 50 km resolution in the corners of the domain (Figure 1). It corresponds to 166 (longitude) × 122 (latitude) horizontal grid cells. The model has 20 vertical layers, from the surface to 200 hPa, with 8 layers within the first two kilometers.

The model is driven by the European Centre for Medium-Range Weather Forecasts (ECMWF) global meteorological fields
(Owens and Hewson, 2018). Both CHIMERE (Menut et al., 2013) and its adjoint code operate the MELCHIOR-2 chemical scheme, with more than 100 reactions (Lattuati, 1997; Derognat et al., 2003), including 24 for inorganic chemistry. Considering the NO$_2$ short lifetime, we do not consider its import from outside the domain: its boundary conditions are set to zero. Nevertheless, the lateral and top boundaries for other species such as ozone O$_3$, nitric acid HNO$_3$, peroxyacetyl nitrate PAN, formaldehyde HCHO, participating to the NO$_x$ chemistry, are considered. Initial and boundary conditions when relevant
are specified using a nested run of CHIMERE (Siour et al., 2013) over a European domain (15.25 °W-35.75 °E; 31.75 °N-74.25 °N) with a spatial resolution of 0.5 °, using itself boundary and initial conditions from climatological values from the LMDZ-INCA global model (Szopa et al., 2009).

The aerosol module of CHIMERE is not considered in the simulations and inversions, as the adjoint of this module is not available (Fortems-Cheiney et al., 2021; Savas et al., 2023).

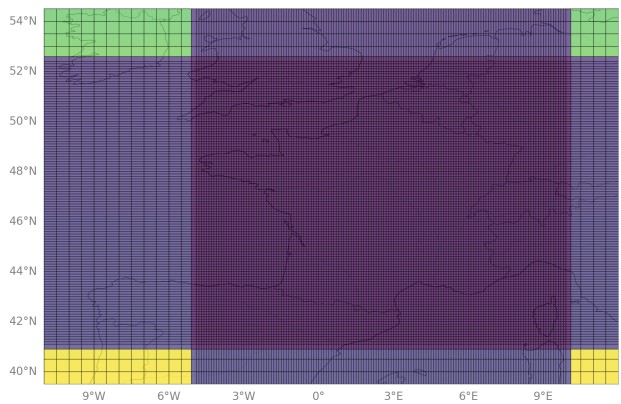

**Figure 1.** *Domain of our CHIMERE configuration: 10 km resolution regular sub-grid in purple, 50 km resolution in green and yellow, 10×50 km$^2$ resolution in blue.*

## 2.3 TROPOMI-PAL observations

TROPOMI, launched onboard Sentinel 5 Precursor in October 2017, is in a near-polar sun-synchronous orbit (approximate altitude of ~824 km) with an ascending node equatorial crossing at ~13h40 Mean Local Solar time. With an orbital cycle of 16 days, and 14 orbits a day, the satellite passes over the same geographical area every cycle of 227 orbits. With a swath as wide as 108 °− approximately 2600 km on ground – TROPOMI provides daily coverage for $NO_2$. Observations over our domain span from around 9:30 am to 2:30 pm local time with data from 2 to 3 orbits per day, with the major coverage around noon (11:30 am to 1:30 pm).

An evaluation of the TROPOMI-v1.3 product with surface remote sensing observations had indicated a systematic low bias of TROPOMI $NO_2$ tropospheric vertical columns densities (TVCDs) of typically -23 % to -37 % in clean to slightly polluted conditions and as high as -51 % over highly polluted areas (Verhoelst et al., 2021) compared to ground-based measurements. This negative bias has been mainly attributed to a negative cloud height bias in the Fast Retrieval Scheme for Clouds from Oxygen absorption band (FRESCO) implementation (van Geffen et al., 2022b) and efforts have been made to correct it in the TROPOMI-PAL product (Eskes et al., 2021).

Here, we use the PAL TROPOMI reprocessed data (Eskes et al., 2021), available from 2019 to the $11^{th}$ of November, 2021. We use a recent reprocessing of the TROPOMI data, called RPRO version 2.4, to cover the end of the year 2021. This latest reprocessing uses a new higher-resolution directional Lambertian-equivalent reflectivity derived from TROPOMI observations (van Geffen et al., 2022a). Nevertheless, the last evaluation of the TROPOMI RPRO v2.4 product with surface remote sensing observations still indicates significant biases of TROPOMI the $NO_2$ TVCDs of typically +13 % over clean areas to -40 % over highly polluted areas (Lambert et al., 2023). Part of theses biases could be explained by the relatively coarse horizontal resolution of the global TM5-MP prior profiles (1 ° × 1 °) used in the retrieval process (which can be neglected when applying the retrieval averaging kernels to the model) (Douros et al., 2023). However, these biases are also largely attributed to systematic errors in the retrieved cloud pressure, surface albedo used etc. (Boersma et al., 2016; Douros et al., 2023).

We select the data with a quality assurance (qa) value of 0.75 for both products, following the criteria of van Geffen et al. (2022b). The latest version of the TROPOMI $NO_2$ Algorithm Theoretical Basis Document (ATBD) is now for product version 2.6 (TROPOMI ATBD of the total and tropospheric $NO_2$ data products, KNMI, S5P-KNMI-L2-0005-RP, issue 2.4.0, van Geffen et al. (2022a)).

## 2.4 Comparison between simulated and observed $NO_2$ TVCDs

To make relevant comparisons between simulations and satellite observations, the averaging kernels (AKs), which are associated with each observed TVCD and representing the vertical sensitivity of the satellite retrieval (Eskes and Boersma, 2003) are applied to the simulated concentration field. Due to the over-sampling resulting from the model's horizontal resolution being coarser than the TROPOMI data, we aggregate spatially and temporally the TROPOMI observations at the CHIMERE resolution into so-called super-observations. Within a given grid cell and time step, the super-observation is the observation (TVCD and AKs) which is the closest to the mean of the TROPOMI TVCDs. The error associated with each super-observation is also

derived from the observation closest to the mean value and subsequently included into the total so-called "observation error" (see Section 2.5). Our derivation of the error associated with each super-observation is thus conservative compared to other studies (Boersma et al., 2016) where the super-observation uncertainty is reduced compared to that of individual observations. The reduction of uncertainty when combining several observations account for the fact that the retrieval errors include random noise (in particular, instrumental noise) without spatial correlation, i.e. errors which are independent from one observation to the other. However, as discussed above, the TROPOMI $NO_2$ observations bear large systematic errors from the retrieval process, which can exhibit significant spatial correlations. This explains our conservative attribution of observation errors to the super-observations.

The corresponding column of $NO_2$ in CHIMERE is vertically interpolated (at TROPOMI's super-observation location) on the vertical levels of the super-observation retrieval, and vertically integrated with the AKs of the super-observation, to yield the $NO_2$ simulated TVCD to be compared to the super-observation TVCD, as illustrated in Figure 2a) and Figure 2b) for the month of April 2020.

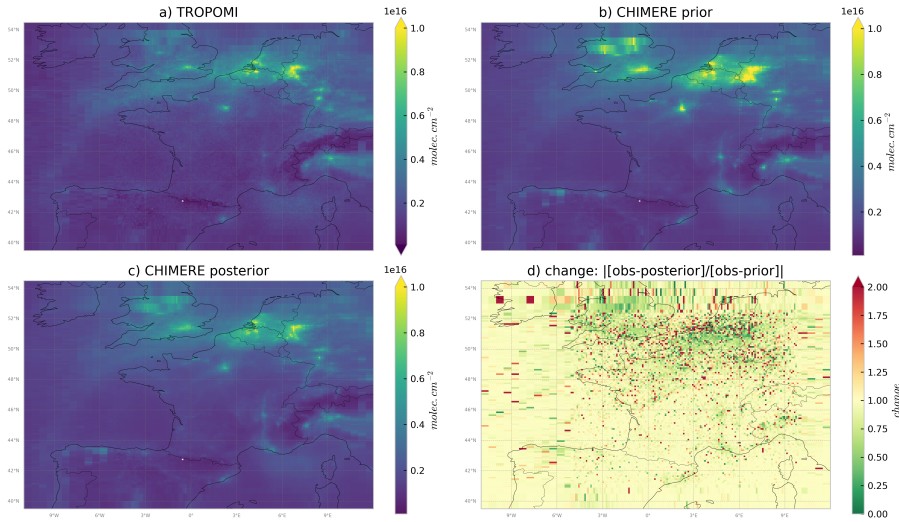

**Figure 2.** *Averages of $NO_2$ TVCDs by a) the TROPOMI-PAL data, b) the CHIMERE simulation using prior emissions from the CAMS-REG/INS and MEGAN inventories, described in Section 2.1 and c) the CHIMERE simulation using the posterior emissions from the inversion, for April 2020, in* $\mathrm{molec.cm^{-2}}$. *d) Ratio of the posterior and prior biases between $NO_2$ TVCDs simulated with CHIMERE and the TROPOMI-PAL observations. All ratios lower than 1, in green, demonstrate that posterior emission estimates improve the simulation compared to the prior ones.*

## 2.5 Variational inversion of $NO_x$ emissions

The inversion of $NO_x$ emissions consists in correcting the prior estimate of the emissions (presented in Section 2.1) to improve the fit between $NO_2$ TROPOMI-PAL satellite data and their simulated equivalents, using a Bayesian variational inversion

framework similar to that of Fortems-Cheiney et al. (2021).

Series of 7-day inversion windows – independent from each other – are run and then combined to provide a corrected ("pos-
terior") estimate of $NO_x$ emissions over the whole period of analysis (2019-2021). For each inversion window, the posterior
estimate of the emissions is found by iteratively minimizing the cost function $J(\mathbf{x})$:

$$J(\mathbf{x}) = (\mathbf{x} - \mathbf{x}^b)^T \mathbf{B}^{-1}(\mathbf{x} - \mathbf{x}^b) + (\mathcal{H}(\mathbf{x}) - \mathbf{y})^T \mathbf{R}^{-1}(\mathcal{H}(\mathbf{x}) - \mathbf{y}) \tag{1}$$

where $\mathbf{x}$, $\mathcal{H}$, $\mathbf{y}$, $\mathbf{B}$, $\mathbf{R}$ are respectively the control vector, the observation operator, the satellite observations, the prior error
covariance matrix and the observation error covariance matrix. The control vector $\mathbf{x}$ gathers variables for the correction of the
surface NO and $NO_2$ emissions, $\mathbf{x}^b$ corresponding to the prior estimate of the control vector.

Here the definition of $\mathbf{x}$ ensures that the inversion solves separately for the two main types of NO emissions: the anthropogenic
and the biogenic emissions (without any further sectorization or decomposition into more detailed emission components), and
for the anthropogenic $NO_2$ emissions, at the 1-day and model grid-cell (i.e. 50 to 10 km) resolution temporally and horizontally,
and over three vertical levels for the anthropogenic emissions (accounting for the range of injection heights discriminating
between the anthropogenic sources). With such a control vector, the prior $NO/NO_2$ anthropogenic emission ratio speciation
from the GENEMIS recommendations (see Section 2.1) is not conserved the inversion, but the analysis focus on the $NO_x$
emissions as the sum of the NO and $NO_2$ emissions.

Furthermore, contrarily to Fortems-Cheiney et al. (2021), we implicitly aim to characterize the prior uncertainty in both the
anthropogenic and biogenic $NO_x$ emissions with log-normal distributions. This allows the inversion system to apply high
variations in $NO_x$ emissions while ensuring positivity, unlike the classic corrections of the emission with scaling factors with a
Gaussian distribution of prior uncertainty. However, the uncertainty in the control vector must follow a Gaussian distribution,
as required by the use of Equation 1. Therefore, here, the control vector $\mathbf{x}$ is defined as the logarithm of the scaling factors
to be applied to the prior estimate of the emissions, and the posterior anthropogenic or biogenic emission estimate at a given
grid-cell of the model and for a given day $\mathbf{f}_i$ is derived from the corresponding control parameter $\mathbf{x}_i$ as $\mathbf{f}_i = \exp(\mathbf{x}_i) . f_i^{prior}$.
Our control vector $\mathbf{x}$ therefore contains:

- the logarithm of the scaling coefficients for NO anthropogenic emissions at a 1-day temporal resolution, for the 166
  (longitude) $\times$ 122 (latitude) horizontal grid cells of the model, and over three vertical bands of injection heights for
  the emissions (from 0 to 25 m, from 25 to 1900 m, from 1900 to 12000 m); this is done essentially to reduce the
  dimensionality of the problem along this axis as most emissions are concentrated in the first 2 layers. The corrections
  applied are the same for all layers within a band,

- the logarithm of the scaling coefficients for $NO_2$ anthropogenic emissions at the same temporal and spatial resolutions
  as for NO,

- the logarithm of the scaling coefficients for NO biogenic emissions at a 1-day temporal resolution, for the 166 (longitude)
  $\times$ 122 (latitude) model grid cells at the surface level.

NO and NO$_2$ 3D initial conditions (specified using a nested run of CHIMERE, as described in Section 2.2) are not controlled and are set once initially for all 7-day windows at 0:00 UTC on the first day of these windows. We therefore do not account for the potential update of the concentrations during a previous 7-day window due to the inversions. They have a low impact on the inversion since the first satellite observations over the domain are around noon while the NO$_x$ lifetime is short (on the order of a few hours, Hakkarainen et al. (2024)).

The uncertainties in the observations $\mathbf{y}$ together with that in the observation operator $\mathcal{H}$, and the uncertainties in the prior estimate of the control vector $\mathbf{x}^b$ are assumed to have a Gaussian distribution. Therefore, they are fully characterized by error covariance matrices.

The prior uncertainty is defined at the resolution of the control vector. Thus, the terms of the prior error covariance matrix $\mathbf{B}$ reflect the uncertainties in the logarithm of the anthropogenic and biogenic emissions of NO$_x$ at 1-day and 50 to 10 km

resolution. This matrix is set block diagonal (see Equation 2), with a block corresponding to the anthropogenic emissions, and the other one corresponding to the biogenic emissions, assuming that there is no correlation between the respective uncertainties in the prior estimates for these two types of emissions.

Another assumption is that at the 1-day and 50 to 10 km resolution, there is no spatial or temporal correlations in the prior uncertainties in the anthropogenic emissions, due to the heterogeneity of these emissions. Therefore, the first block of $\mathbf{B}$

corresponding to the logarithms of the anthropogenic emissions is set diagonal. Each diagonal element is set at $(0.3)^2$: the range associated to this $\sigma$ value in the log-space corresponds to a factor ranging between 74 %-135 % in the emission space at 1 day and pixel at model's grid scale.

The second block of $\mathbf{B}$ corresponding to the biogenic fluxes accounts for space correlations in the uncertainties in these emissions, which are assumed to be more homogeneous in space and time han the anthropogenic emissions, and to decrease

exponentially with distance. They are set with a $\lambda_0 = 30$ km decorrelation length. On the diagonal, uncertainties are set to a value of $(0.6)^2$. The range associated to this value $\sigma$ in the log-space corresponds to a factor ranging between 55 %-182 % in the emission space at the 1-day and model's grid scale.

$$
\mathbf{B} =
\begin{pmatrix}
\begin{pmatrix}
\sigma_{ii}^{ant} & \cdots & 0 \\
\vdots & \ddots & \vdots \\
0 & \cdots & \sigma_{jj}^{ant}
\end{pmatrix} & \mathbf{0} \\
\mathbf{0} &
\begin{pmatrix}
\sigma_{j+1,j+1}^{bio} & \cdots & \nu_{k,j+1}^{bio} \\
\vdots & \ddots & \vdots \\
\nu_{j+1,k}^{bio} & \cdots & \sigma_{kk}^{bio}
\end{pmatrix}
\end{pmatrix}
\tag{2}
$$

with $\nu$ of the form: $\nu_{nm} = \sigma_n \sigma_m e^{\frac{-\Delta x}{\lambda_0}}$

$\mathcal{H}$ is the observation operator, linking the control variables in the log-space to the simulated equivalents of the super-observations. It includes the exponential operator and scaling factor converting the maps of the logarithm of the coefficient for the emissions

into emission maps at the spatial and temporal resolutions of CHIMERE, the atmospheric chemistry and transport model CHIMERE itself, and the extraction of the TVCDs from CHIMERE where and when we have TROPOMI super-observations.

The uncertainties on the observations y and on the observation operator are characterized by the so-called observation error covariance matrix $\mathbf{R}$, set-up here as a diagonal matrix based on the assumption that these errors are not correlated in space or time when aggregated at the model 50 to 10 km and 1-hour resolution. The variance of the observation errors corresponding to individual observations in the diagonal of $\mathbf{R}$, is the quadratic sum of the error we have assigned to the TROPOMI-PAL super-observations (see Section 2.4), and of an estimate of the errors from the observation operator. We assume that the observation

operator error is dominated by the chemistry-transport modeling errors and by the errors associated with the discrepancies between the spatial representativity of the super-observations and of the model corresponding column: it is set at 30 % of the retrieval value. It was set at 20 % by Fortems-Cheiney et al. (2021) at a coarser resolution and is increased here to take into account the mismatch between the shape and location of the real and simulated atmospheric patterns at our finer resolution (see Section 2.4).

The minimum of the cost function $J$ is searched for with the iterative M1QN3 limited-memory quasi-Newton minimization algorithm (Gilbert and Lemaréchal, 1989). At each iteration, the computation of the gradient of $J$ relies on the adjoint of the observation operator, and in particular on the adjoint of CHIMERE. In the results presented in Section 3, as a compromise between computational time and the level of convergence of the iterative minimization of $J$ in the inversions, the minimization is considered to be satisfying when the norm of the gradient of $J$ is reduced by 80 %.

The calculation of the uncertainty in the posterior estimates of emissions is challenging when using a variational inverse system (Kadygrov et al., 2015; Rayner et al., 2019; Fortems-Cheiney et al., 2021) and it is not done here. It would require a large ensemble of time-consuming inversions (especially due to the handling of chemistry), to enable a proper sampling of the uncertainties and thus a proper derivation of the of the statistics of uncertainty.

## 3    Results

### 3.1    Seasonal and inter-annual estimates of NO$_x$ total French emissions

#### 3.1.1    Fit between TROPOMI-PAL super-observations and their simulated equivalents

Before analyzing the results in terms of emissions, we check the behavior of the inversion by comparing the performances of the prior and posterior simulations in reproducing the spatial and temporal variations of the observations. As an illustration, TROPOMI-PAL and the corresponding CHIMERE NO$_2$ TVCDs are shown in 2a and Figure 2b, respectively, for April 2020. The TROPOMI-PAL observations and their NO$_2$ simulated equivalents present similar spatial patterns, with hotspots (TVCDs higher than $1\times10^{16}$ molec.cm$^{-2}$) over urban areas and low values over rural ones during the whole simulated period from 2019 to 2021 (illustrated in Figure 2a) and b) for one month in 2020). However, the prior simulation overestimates the NO$_2$ TVCDs over urban areas in France compared to the observations. For example, for April 2020, the mean bias between the prior simulation and TROPOMI is of about $4.2\times10^{14}$, $2.0\times10^{14}$ and $3.8\times10^{14}$ molec.cm$^{-2}$ over Paris, Lyon and Marseille (Figure A1), respectively. The inversion brings the simulated TVCDs closer to the TROPOMI-PAL data over urban areas (Figure 2c): in this case, the mean bias and the mean RMS over the three cities are reduced by about 30 % and 12 % respectively.

As the CHIMERE prior simulation overestimates the NO$_2$ TVCDs, the inversion brings the CHIMERE NO$_2$ columns closer to the TROPOMI-PAL data by reducing NO$_x$ emissions, mainly over dense urban areas (Ile-de-France, Lyon-Marseille axis, London area, Benelux, Frankfurt, Po Valley, see Figure 3b). Over these areas, the relative corrections provided by the inversions to the total (anthropogenic and biogenic) emissions can reach -70 % (Figure 3b).

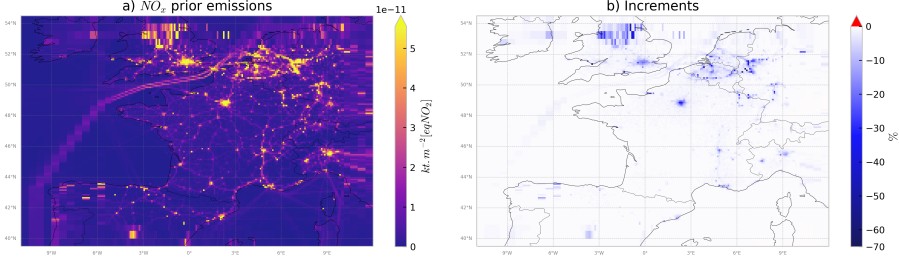

**Figure 3.** *a) NO$_x$ total prior fluxes (anthropogenic emissions from CAMS-REG/INS and biogenic emissions from MEGAN, see Section 2.1 for details) and b) relative increments to the prior total emissions from the inversion in %, for April 2020.*

#### 3.1.2    Estimates of NO$_x$ French total emissions

This section focuses on the results from the NO$_x$ inversions in terms of comparisons between the total posterior NO$_x$ emissions and the prior ones. The focus on total emissions is explained by the fact that the distinction between the signal from the biogenic and anthropogenic emissions in the comparisons between the chemistry transport model and the satellite NO$_2$ observations is challenging in the inversion framework. Since both types of emissions are solved at the 1-day and model grid cell resolution,

the inversion system relies on the respective amplitude and spatial correlations in the prior uncertainties in the biogenic and anthropogenic emissions (in $\mathbf{B}$) to discriminate between the corrections to be applied to the prior estimates of the two types of emissions. However, our configuration of $\mathbf{B}$ yield similar structures of spatial correlations for the two types of emissions. Therefore, our confidence in the split between biogenic and anthropogenic emissions in the posterior emission estimates is low whenever the two types of emissions have comparable levels according to the prior emission estimates.

At the national scale, both the prior and the posterior emissions present a similar seasonal cycle in 2019-2021, with monthly emissions higher than 72 $\mathrm{kteqNO_2}$ during winter and equal to or lower than 66 $\mathrm{kteqNO_2}$ during summer (Figure 4).

The mean French national budget for the years 2019 to 2021 from the posterior total emission estimates is about 850 $\mathrm{kteqNO_2}$, which is lower than the total prior emissions estimated from the CAMS-REG/INS and MEGAN inventories (average of 875 $\mathrm{kteqNO_2}$, Figure 3, Table 1), with the largest reductions reaching about 8 % during fall and winter. This is expected as public policies have led to regular reductions in the anthropogenic emissions between 2016 (the year of our CAMS-REG/INS inventory) to 2019-2021. The decrease in total $\mathrm{NO_x}$ emissions is estimated at -13 % from 2016 to 2019 by the French Technical Reference Center for Air Pollution and Climate Change (CITEPA report, we use the estimates for "out-of-scope" natural emissions which are not reported to UNFCCC). According to the CITEPA, the reduction is driven by large reductions in emissions from three major sectors: the energy sector (-28 %), the industry sector (-22 %), and the transport sector (-15 %) between 2016 and 2019. However, the decrease found in this work is only about -3 % (Table 1). This can be explained by our configuration with neither spatial nor temporal correlation in our $\mathbf{B}$ matrix, leading to null or very small correction of the prior emissions from the inversions when the coverage of the country by TROPOMI super-observations is very sparse (Zheng et al., 2020). In this case, the posterior emissions remain close to the prior emission estimate and therefore, at their 2016 level (see Section 3.2.3).

The 2019-2021 inter-annual variability is smaller with our inversions (Table 1) than in the estimates from the CITEPA: the annual budget of the French total $\mathrm{NO_x}$ posterior emissions varies by less than 1 % from year-to-year, while the CITEPA estimates a decrease of about 13 % in 2020 compared to 2019, and an increase of about 3 % in 2021 compared to 2020 (Table 1). The similar annual total emissions in 2019 and 2020 nevertheless overlay different sub-annual variations. Higher emissions — partly associated with higher TROPOMI $\mathrm{NO_2}$ tropospheric columns (not shown) — are estimated in January, in February, in June and in November 2020 compared to 2019 (Figure 4). These increases counterbalance the decrease of $\mathrm{NO_x}$ emissions in March/April 2020 (Figure 4, Figure 5) which could be both associated with the COVID-19 crisis and to meteorological conditions with a warmer winter in 2020 than in 2019 (see Section 3.2.2).

When considering the split between biogenic and anthropogenic emissions from the inversions despite its lack of reliability, the posterior biogenic emission estimates are close to the prior estimates derived from MEGAN (see Figures C1 and C2 in supplement): in particular they do not appear to be stronger in 2020 than in 2019. Therefore, the low decrease (-3 %) of the posterior estimate of the national budget of the total $\mathrm{NO_x}$ emission in 2020 can hardly be explained by an increase in the biogenic emissions which would compensate for the decrease of the anthropogenic emissions, even though the biogenic emissions due to agriculture have been overlooked in our inversion set-up (see Section 2.1).

| Domain or region | Period | CAMS-REG/INS + MEGAN inventories | Posterior in 2019 | | Posterior in 2020 | | Posterior in 2021 | | Posterior 2020-2019 | CITEPA 2020-2019 | Posterior 2021-2020 | CITEPA 2021-2020 |
|---|---|---|---|---|---|---|---|---|---|---|---|---|
| | | kteqNO$_2$ | kteqNO$_2$ | [%] | kteqNO$_2$ | [%] | kteqNO$_2$ | [%] | [%] | [%] | [%] | [%] |
| France | Annual | 875 | 853 | -3 | 852 | -3 | 845 | -3 | 0 | -13 | -1 | +3 |
| | Spring (MAM) | 227 | 224 | -1 | 222 | -2 | 222 | -2 | -1 | -21 | 0 | +12 |
| | March | 78 | 76 | -3 | 74 | -5 | 75 | -4 | -2 | -12 | +1 | +5 |
| | April | 77 | 76 | -1 | 75 | -4 | 74 | -3 | -2 | -30 | 0 | +23 |
| | May | 73 | 72 | -1 | 73 | -1 | 72 | -1 | +2 | -22 | -2 | +10 |
| | November | 69 | 65 | -6 | 67 | -4 | 66 | -4 | +4 | -19 | -2 | +14 |

**Table 1.** *Prior and posterior NO$_x$ total emission budgets in* kteqNO$_2$ *and their relative differences in %* $[100 \times (posterior - prior)/prior]$, *in France for different periods. Columns "Posterior* year$_n$ $-$ year$_{n-1}$*" and "CITEPA* year$_n$ $-$ year$_{n-1}$*" show the relative difference between* year$_n$ *and* year$_{n-1}$ *posterior fluxes and CITEPA in %* $[100 \times (F_{year_n} - F_{year_{n-1}})/F_{year_{n-1}}]$.

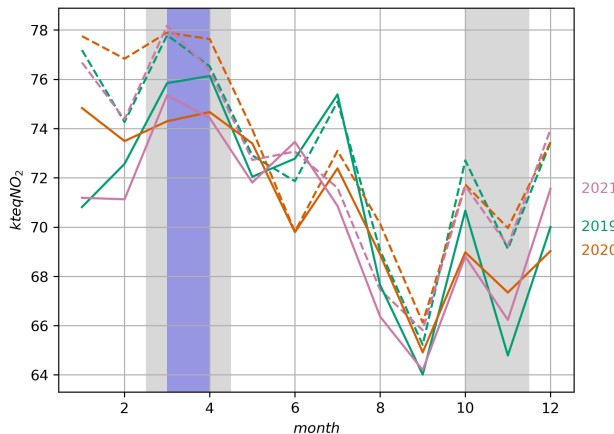

**Figure 4.** *Monthly NO$_x$ total emissions in France as estimated by the CAMS-REG/INS and MEGAN inventories (dotted lines) and by the inversions for years 2019 (in green), 2020 (in orange) and 2021 (in pink), in* kteqNO$_2$.month$^{-1}$. *Grey shaded areas show the French lockdown periods for the year 2020 and the purple shaded area shows the French lockdown period for the year 2021.*

## 3.2 Impact of the COVID-19 lockdown in spring 2020

The atmospheric lifetime of NO$_2$ dictates that the high spatial resolution measurements from TROPOMI should readily capture rapid week-to-week changes in near-surface emissions (Levelt et al., 2022) and should therefore make it possible to assess the impact of the COVID-19 lockdown in spring 2020 on NO$_x$ French emissions. Following a usual diagnostic in the literature to assess the change in air pollutant concentrations due to the COVID-19 policies, we characterize the impact of the first COVID-

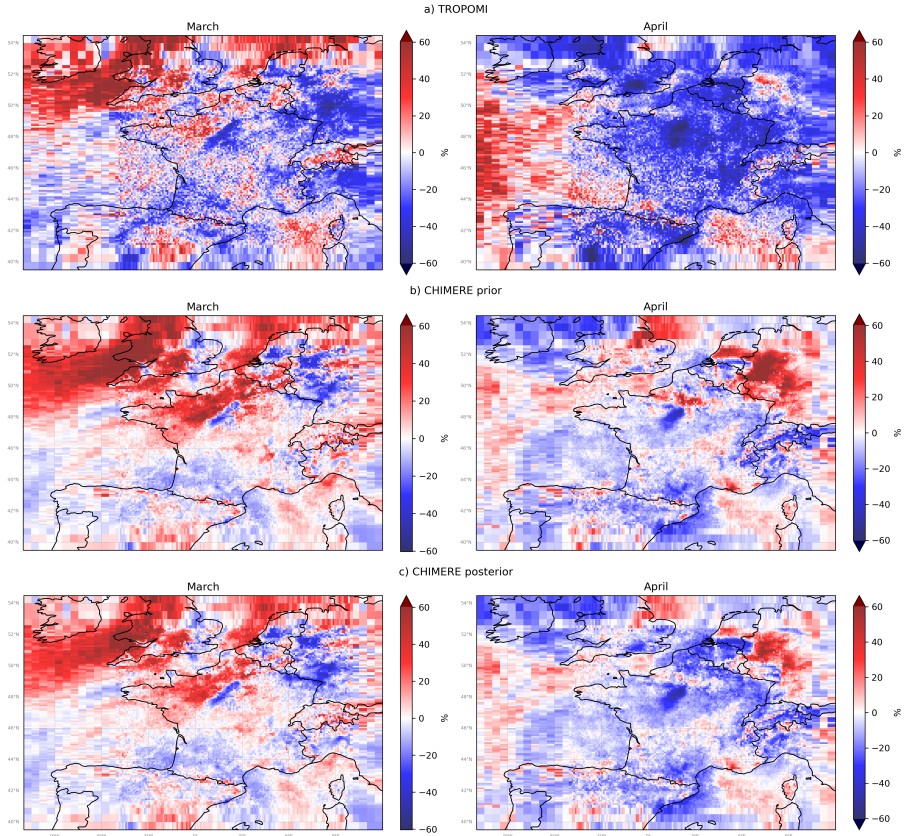

**Figure 5.** *Monthly gridded relative differences between monthly averages of a) TROPOMI-PAL NO$_2$ tropospheric columns, b) CHIMERE prior tropospheric columns and c) CHIMERE posterior tropospheric columns estimated by the inversions from March/April 2019 to March/April 2020, in % $[100 \times (F_{2020} - F_{2019})/F_{2019}]$.*

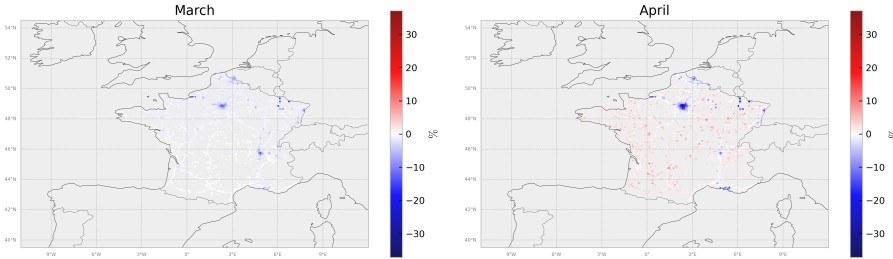

**Figure 6.** *Monthly gridded relative differences between the monthly total (anthropogenic + biogenic) posterior emissions estimated by the inversions from March/April 2019 to March/April 2020, for the months of March/April, using an urban and road land-use proxy (see Figure C3), in % $[100 \times (F_{2020} - F_{2019})/F_{2019}]$.*

19 lockdown in France — occurring from March $17^{th}$ to May $10^{th}$ 2020 — in terms of changes from spring 2019 to spring 2020. Our analysis relies on the spatial distribution of the main areas of anthropogenic activities to identify the locations where the total $NO_x$ emissions should be dominated by anthropogenic emissions.

### 3.2.1 General impact on $NO_2$ TROPOMI TVCDs

The TROPOMI $NO_2$ TCVDs in March/April 2020 are first compared to 2019 at the national scale (Table 2). The TROPOMI
$NO_2$ TCVDs decrease from 2019 to 2020 by -11 and -28 %, respectively in March/April over France. Similar decreases have been diagnosed in the measurements at surface stations over Europe with reductions in the average concentrations of $NO_2$ of about -25 % for at least 75 % of the 1308 European Air Quality e-Reporting database (AirBase) stations compared with the average of the previous seven years (2013–2019) for the period March 18–May 18, 2020 (Deroubaix et al., 2021).

The population distribution is heterogeneous in France, with large rural areas (i.e., 75.7 % of the country area, according to
375 the French National Institute of Statistics and Economic Studies, 2020). We focus here on 8 urban areas which correspond to hotspots of $NO_x$ concentrations, and where, as a consequence, we expect a stronger signal due to changes in anthropogenic activities (Figure A1): Paris, Lyon, Marseille, Lille, Bordeaux, Toulouse, Nice and Nantes.

The strength of the TROPOMI $NO_2$ signal differs over these 8 French cities. Indeed, the absolute changes in $NO_2$ TVCD in March/April 2020 compared to March/April 2019 are higher for Paris, Lille, Lyon and Nantes (see Figure C5 in Supplemen-
380 tary materials) than for Bordeaux, Marseille, Nice and Toulouse (i.e., $1.\times10^{15}$ molec.cm$^{-2}$ or less). The population density difference between those cities could partly explain such variability (Table A1).

TROPOMI $NO_2$ tropospheric columns in March/April 2020 are 26 to 38 % lower on average than in 2019 over the 8 urban areas (Table 2). The relative changes from April 2019 to April 2020 range from -54 % for Paris to -27 % in Bordeaux (Table 2). This relative change over Paris is consistent with the decrease of -52 % described by (Levelt et al., 2022) and with the decrease
of about -56 % estimated by the tropospheric $NO_2$ columns measured by two UV-Visible Système d'Analyse par Observation Zénithale instruments (SAOZ, Pazmiño et al. (2021)).

The temporal variability of the changes from spring 2019 to spring 2020 also differs from one urban area to another. Excepted for Bordeaux and Toulouse, the reductions of TVCDs are higher in April 2020 (Figure 6, Table 2) than in March 2020 (Figure 5, Table 2). This is consistent with the fact that the French population has been confined only from mid-March (i.e., on March
$17^{th}$) versus the whole of April in 2020.

### 3.2.2 Various factors contributing to the differences in concentrations from spring 2019 to spring 2020 beyond the lockdown

The relative changes in the TROPOMI TVCDs from spring 2019 to spring 2020 are partly due to COVID-19 lockdowns but they are also driven by the changes in the meteorological and atmospheric chemistry transport conditions (Menut et al., 2020;
Diamond and Wood, 2020; Petetin et al., 2020; Barré et al., 2021; Gaubert et al., 2021; Deroubaix et al., 2021).

The differences from spring 2019 to spring 2020 in the simulated CHIMERE prior $NO_2$ TVCDs (Figure 5) are mainly due to changes in the meteorology and to a lesser extent to changes in the model boundary conditions and in the biogenic emissions

| Urban area | TROPOMI TVCD | | CHIMERE prior TVCD | | CHIMERE posterior TVCD | | Emissions posterior | |
|---|---|---|---|---|---|---|---|---|
| | Mar | Apr | Mar | Apr | Mar | Apr | Mar | Apr |
| France | -11 | -28 | +7 | -4 | +2 | -12 | -2 | -1 |
| Bordeaux | -34 | -27 | -14 | +23 | -15 | +21 | -3 | -1 |
| Lille | -15 | -35 | +19 | -10 | +4 | -28 | -9 | -11 |
| Lyon | -27 | -43 | +13 | +18 | +1 | +1 | -15 | -13 |
| Marseille | -20 | -27 | -2 | +46 | -9 | +19 | -2 | -7 |
| Nantes | +2 | -45 | +16 | -2 | +14 | -7 | -3 | -3 |
| Nice | -28 | -37 | 0 | +7 | -3 | +2 | 0 | -1 |
| Paris | -41 | -54 | -4 | +42 | -12 | -9 | -11 | -26 |
| Toulouse | -41 | -31 | -23 | -23 | -22 | -22 | 0 | +1 |

**Table 2.** *Changes in NO$_2$ TROPOMI-PAL and CHIMERE tropospheric columns (in %) and changes in NO$_x$ total posterior emissions (in %), between March/April 2019 and March/April 2020, for the 8 French urban areas displayed in Figure A1.*

(Section 2.2), as the prior anthropogenic emissions are the same in 2019 and in 2020 (Section 2.1).

CHIMERE prior NO$_2$ TVCDs are 23 % higher in March 2020 than in March 2019 in the Northern part of France, excepted for the plume of Paris. These changes have thus a sign which is opposite to the changes expected from the COVID-19 lockdowns. This is in agreement with the analysis of Gaubert et al. (2021), who have shown that when considering only the effect of meteorological variability, the level of NO$_2$ concentrations would have been high during the 15 March–14 April 2020 period compared to the average NO$_2$ concentrations at the same period over the five previous years (2015–2019) in the north-western part of France (i.e., Bretagne, Pays de la Loire, Normandie and Hauts-de-France regions, Figure 5).

CHIMERE prior NO$_2$ TVCDs are 4 % lower in April 2020 than in April 2019 over almost the entire country (excepted above Paris and in the Rhone Valley). This is consistent with temperatures above seasonal values (by 3 °C over France, ranking as the third warmest April on record) and to persistent anticyclonic conditions in April 2020 (MeteoFrance, 2022)).

CHIMERE posterior NO$_2$ TCVDs are about 12 % lower in April 2020 than in April 2019 over almost the entire country (Figure 5, Table 2); a focus at the city scale shows a decrease of about 9 % over Paris between April 2020 and April 2019 whereas the prior NO$_2$ TCVDs increase by about 42 % (Table 2).

Finally, the meteorological conditions also have an impact on the availability of the satellite observations. Due to the more favorable meteorological conditions with less cloud coverage leading to an unusual clear sky period in April 2020 (Gaubert et al., 2021; Deroubaix et al., 2021), there is a higher number of TROPOMI observations in April 2020 than in March 2020 and than in 2019, particularly over the northeastern part of France (Figure C4 in Supplementary materials), that may allow for a better correction of the CHIMERE NO$_2$ TCVDs for this particular period (Section 3.2.3).

### 3.2.3 Impact on NO$_x$ anthropogenic emissions from spring 2019 to spring 2020

At the French national scale, the total NO$_x$ emission estimates from the inversions present their largest decrease during the first lockdown in March and April 2020 compared to 2019 (Figure 4). The emissions between May and September are similar in 2020 to 2019 (Figure 4), coinciding with the ease of restrictions. Nevertheless, the decrease of NO$_x$ emissions from March/April 2019 to March/April 2020, of about -2 % and -3 % respectively at the national scale, is relatively flatter than the estimations found in the literature. For example, the CITEPA estimates reductions of about -12 and -30 % (Table 1), respectively, from March/April 2019 to March/April 2020 at the French national scale. Meteorology with a warmer winter can explain part of the changes in emissions between 2019 and 2020 for business sectors such as the residential combustion (Barré et al., 2021; Guevara et al., 2023).

We thus analyze the impact of the COVID-19 policies in terms of differences in retrieved anthropogenic emission estimates from the inversion, from spring 2019 to spring 2020. For this, we focus on urban areas, assuming that the emissions in these pixels (see Figure A1 for the chosen locations) are almost entirely due to anthropogenic activities.

In our inversions, the changes are negative in 7 of the 8 chosen urban areas (Figure 6, Table 2), qualitatively consistent with the reduction in the intensity of vehicle traffic (Guevara et al., 2021, 2022). The changes are also negative for urban areas outside France in our domain (Table C4). The impact of the lockdown on NO$_x$ anthropogenic emissions is very different from one urban area to another. These differences between cities cannot be explained by different contributions of industry to NO$_x$ emissions in or around these urban areas as INS data do not show major differences in terms of sectoral distribution between the main French cities.

The highest reductions are seen in Paris with about -26 % in April 2020 compared to April 2019, followed by Lyon (-13 %), and Lille (-11 %, Table 2). Several urban areas only present a very small drop of emissions in spring 2020 (Table 2, Figure 7) e.g. the French urban areas Bordeaux, Nice and Nantes. The Toulouse urban area even show a slight increase (+1 %) in April 2020 compared to April 2019 (Table 2, Figure 7).

These small changes of emissions over Bordeaux, Nice and Toulouse does not seem consistent with the drops in traffic activity estimated by the Centre d'études et d'expertise sur les risques, l'environnement, la mobilité et l'aménagement CEREMA (CEREMA, 2023). A dedicated study might be necessary to understand in details the inter-urban variability and is not done here as up-to-date local inventories are not available for every city, particularly smaller ones.

The reductions in the NO$_x$ anthropogenic emissions from spring 2019 to spring 2020, while substantial, do not exhibit the same magnitude as the reductions in TROPOMI-PAL NO$_2$ TVCDs. This is expected because of the non-linearities between NO$_x$ anthropogenic emissions and NO$_2$ TVCDs but also, due to the limitations and assumptions of the inversion itself.

### 3.2.4 Exploring limitations in our analysis

The discrepancy between the emission changes from 2019 to 2020 between the inversion results and independent estimates from inventories (i.e., CITEPA, CEREMA) can be partly connected to our conservative characterization of the uncertainties in the prior emission estimates, which ignores potential spatial and temporal correlations in the prior uncertainties in the anthro-

pogenic emissions at the daily and 10 to 50 km resolution. Therefore, the direct information from the satellite observations is not extrapolated by the inversion in space and time via correlations in the $\mathbf{B}$ matrix, and thus the departure of the inversion from the prior emission estimates is highly sensitive to the observation coverage and to their footprints in the emission field.

In this context, the corrections provided by the inversions to the prior emissions can be limited by the fact that the potential of TROPOMI to provide information is hampered by the cloud coverage. When considering the annual to monthly budgets of the emissions over all days (with and without observations), the amplitude of the corrections to the prior estimate of the emissions driven by the satellite observations is artificially decreased by the lack of corrections during days when there is no satellite observations.

We quantify this effect by selecting days when we have at least one super-observations over the pixel of the 8 urban areas of interest and extrapolating the retrieved emissions for this subset of days to the whole year, hereafter called "filtered emissions". The reductions in the $NO_x$ anthropogenic filtered emissions from spring 2019 to spring 2020 are almost always higher than for the standard posterior emissions (e.g., -28 % and -24 % over Paris and over Lyon respectively for filtered emissions versus -26 % and -13 %, Table 3). These results with such a focus on days with the best coverage are in principle much closer to the changes in emissions estimated by the CITEPA.

The corrections provided by the inversions to the prior emissions are also highly dependent on the errors associated with the TROPOMI-PAL observations and with the CTM errors in $\mathbf{R}$. We illustrate this effect by performing an inversion without model errors in the covariance matrix $\mathbf{R}$, giving more weight to the satellite data and considering the model as perfect. The posterior $NO_x$ emissions retrieved with this error set-up at the city scale show a higher reduction from spring 2019 to spring 2020 (Tables 3, B3), e.g., -31 % and -25 % over Paris and over Lyon when using all the available observations and -33 % and -42 % for filtered emissions compared to -26 % and -13 % for the reference emissions.

Nevertheless, several urban areas still present a very small drop of emissions even when selecting days with observation and considering the model as perfect to give more weight to the TROPOMI data. This is the case of the French urban areas Bordeaux, Nice and Toulouse with reductions from spring 2019 to spring 2020 lower than 10 % (Table 3).

This may be explained by the strength of the TROPOMI signal over these cities (see Section 3.2.1).

Filtered cases are only indicative as this approach necessitates extrapolating from a notably small sample (days with observations, 16 and 19 days in average for the studied French cities in March and April 2020).

To gain in emission representativity, we suggest exploring a fine tuning of the correlation in $\mathbf{B}$ (with optionally additional information on $NO_2$ concentrations like surface stations) to compensate when TROPOMI data availability is lower. Despite our hypothesis made in Section 2.5 on anthropogenic fluxes, this might be necessary in addressing national-wide $NO_x$ emission monitoring with a high local resolution.

| Urban area | Standard posterior emissions | | Filtered posterior emissions | | Standard posterior emissions $\mathbf{R}$-perfect-model | | Filtered posterior emissions $\mathbf{R}$-perfect-model | |
|---|---|---|---|---|---|---|---|---|
| | Mar | Apr | Mar | Apr | Mar | Apr | Mar | Apr |
| Bordeaux | -3 | -1 | -9 | -2 | -5 | -4 | -13 | -7 |
| Lille | -9 | -11 | -16 | -11 | -10 | -15 | -17 | -13 |
| Lyon | -15 | -13 | -24 | -24 | -15 | -25 | -22 | -41 |
| Marseille | -2 | -7 | -5 | -9 | -2 | -10 | -5 | -13 |
| Nantes | -3 | -3 | -2 | -7 | -5 | -5 | -6 | -11 |
| Nice | 0 | -1 | -4 | -3 | -1 | -4 | -5 | -7 |
| Paris | -11 | -26 | -15 | -28 | -13 | -31 | -15 | -32 |
| Toulouse | 0 | +1 | -1 | +1 | -1 | +1 | -4 | -1 |

**Table 3.** *Changes in $NO_x$ CHIMERE total posterior emissions for the standard (in %) and with a simplified observation error set-up (in %), $\mathbf{R}$ matrix without model error, see details in Section 3.2.3 from March/April 2019 to March/April 2020 considering different TROPOMI coverage for the 8 French urban areas displayed in Figure A1.*

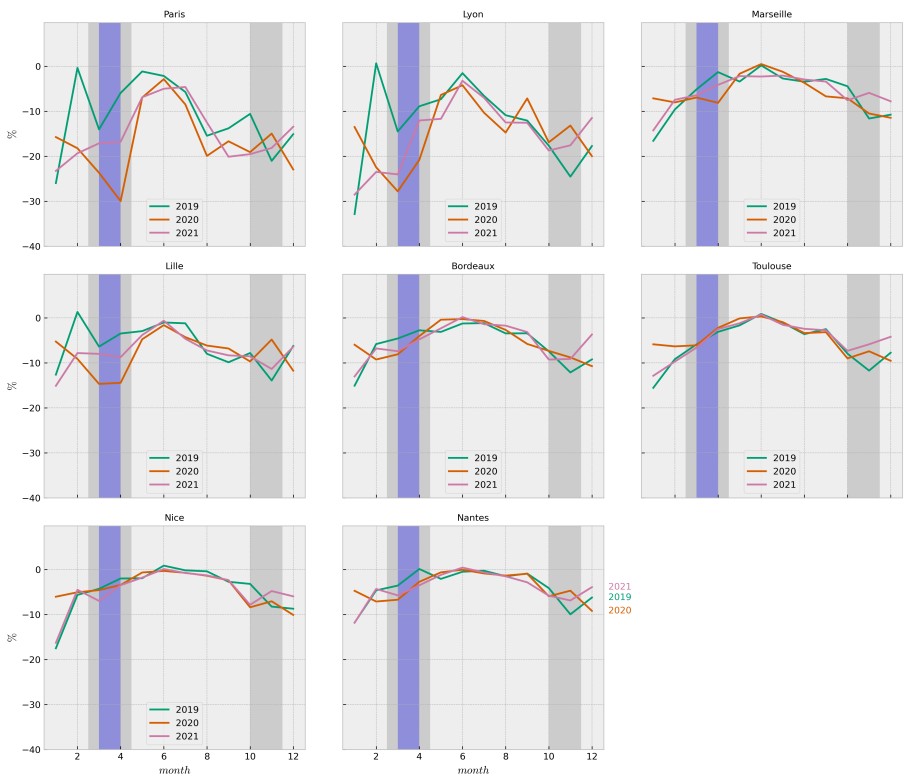

**Figure 7.** *NO$_x$ anthropogenic emissions monthly budget relative differences between posterior and prior for the 8 French urban areas displayed in Figure A1, from inversion results for years 2019 (in green), 2020 (in orange) and 2021 (in pink), in %. Grey shaded areas show the French lockdowns periods for the year 2020 and the purple shaded area shows the French lockdown period for the year 2021.*

## 4 Conclusions

We performed a three-year variational inversion of $NO_x$ emissions from 2019 to 2021 in France at the high resolution of $10 \times 10$ km$^2$. The TROPOMI-PAL observations were assimilated within the inversion system CIF driving the regional CTM CHIMERE with the MELCHIOR-2 chemical scheme.

The French budgets for the years 2019, 2020 and 2021 are about 850 kteqNO$_2$. As expected from the implementation of public policies leading to regular reductions in emissions, these national budgets from the inversions are lower than the CAMS-REG/INS inventory for 2016, used as prior in our inversions. In particular, 2020 does not show a clear reduction in emission compared to 2019: 2020 emissions are higher compared to 2019 in January, February, June, and November. These increases dampen the decline in $NO_x$ emissions in March/April 2020 compared to March/April 2019.

We focus on the changes in $NO_2$ TROPOMI-PAL TVCDs and in $NO_x$ anthropogenic emissions in March and April 2020 compared to 2019, that are due to the lockdowns during the COVID-19 pandemic and to meteorological conditions, including a milder winter (Barré et al., 2021; Guevara et al., 2023). Since inversions mainly detect changes in cities (hotspots), the main changes between March/April 2019 and March/April 2020 are observed at the city scale. However, the impacts of restrictions can vary significantly between different urban areas. Among the 8 selected French urban areas, the relative changes between April 2020 and April 2019 in the TROPOMI-PAL observations ranges from -54 % for Paris to -27 % in Bordeaux. The highest reductions of $NO_x$ anthropogenic emissions, when we focus on days with observations, are seen in Paris with about -28 % in April 2020 compared to April 2019, followed by Lyon (-24 %), and Lille (-11 %).

Several urban areas such as Bordeaux, Nice, and Toulouse, only show a small decrease in emissions from spring 2019 to spring 2020, even when focusing on the results from the inversion over days with at least one super observation and even when more weight is given to the TROPOMI data in the inversion process by assuming that the model is perfect. It may be due to weaker TROPOMI signals over these cities, with $NO_2$ TVCD changes in March/April 2020 compared to March/April 2019 being smaller than those observed in Paris, Lyon, and Lille.

The inversion results show significant decreases in the emissions from the largest cities in France from 2019 to 2020, but this decrease remains lower than that documented in most studies on this topic and by independent inventories (i.e., CITEPA, CEREMA). This can be partly connected to our conservative characterization of the prior emissions uncertainties, since we ignore potential spatial and temporal correlations in the prior uncertainties in the anthropogenic emissions, so that the departure from the prior emission estimates in the inversion are highly sensitive to the observation coverage and emission footprints. The corrections provided by the inversions to the prior emissions can indeed be limited by the cloud coverage affecting the TROPOMI observations, and by errors in the TROPOMI data and in the CTM. Notably, when TROPOMI observations are unavailable, the correction of prior emissions in the unconstrained pixels through inversions is null, resulting in posterior emissions remaining close to the prior ones. Hence, the aggregation of posterior emissions into monthly or yearly budgets leads to a dampening of the signal provided by TROPOMI. In order to better emphasize the direct information from the satellite observations, some of our analysis of the local urban emissions are focused on days with at least one observation in the targeted pixels, yielding a characterization of the COVID-19 effects which is more consistent with the changes in emissions estimated

by the CITEPA.

To explore the impact of the set-up of error covariance matrices, we considered inversions without error models in the covariance matrix $\mathbf{R}$, giving more weight to satellite data. In this case, the posterior $NO_x$ emissions at the city scale exhibit their highest reductions between spring 2020 and spring 2019 (e.g., -31 % and -25 % over Paris and over Lyon respectively, considering all days within the month. Assimilating the observations without accounting for the model errors may lead to over-fitting and thus project these errors onto the emission estimates. However, the weight of $\mathbf{R}$ in our inversions may have to be re-assessed with regards to the relatively conservative option that we use here to assign observation error to the super-observations. A finer assessment would require a good knowledge of the share of retrieval errors between random noise without spatial correlations and more systematic errors with spatial correlations, as well as the typical length scales of such spatial correlations, which is currently challenging to derive (Miyazaki et al., 2012; Boersma et al., 2016; Lambert et al., 2023).

In the absence of alternative information, like new measurements from benchmark cities, reconciling our results with the existing literature would prompt a re-evaluation of the observation errors in our inversion and a potential reassessment of our model error definition. The information contained in TROPOMI TCVDs cannot be fully exploited in our inversion set-up to get constraints on diffuse emissions e.g. in French rural areas.

For improving diagnostics of monthly to annual emissions both at national scale for emission hotspots, addressing challenges arising from satellite coverage gaps involves introducing horizontal, temporal, and sectoral correlations in the covariance matrix $\mathbf{B}$ of the uncertainty in the gridded inventories with hourly variations that are used as prior estimate of the emissions by the inversions. Such a characterization would support the extrapolation in space and time of the information obtained locally for some days from the satellite observation. However, obtaining such correlations at high-resolution poses a substantial challenge. The usual correlation models based on assumptions of isotropy, homogeneity in space and time, and of decrease as a function of distance and time should poorly match the actual derivation and structures of gridded inventories convolved with typical temporal cycles at diurnal to seasonal scales, which is why a conservative configuration was used for the $\mathbf{B}$ matrix in this study (Super et al., 2020). The challenge is exacerbated when tackling a period such as 2019-2021, with lock-down measures in response to the COVID-19 crisis highly impacting the emissions and thus the structures of uncertainties in the emission inventories over large spatial scales but limited periods. Exploring corrections to parameters underlying inventories, such as Fossil Fuel Data Assimilation System (FFDAS) for $CO_2$, may actually support a cleaner extrapolation. Due to the current lack of knowledge about the statistics of the uncertainties in the gridded inventories used for the inversions, a stepwise approach is probably needed to tackle this general problem, including gradually some temporally varying spatial and temporal correlations in $\mathbf{B}$ and, in parallel, increasing efforts to diagnose these uncertainties. If achieved, these additions would assist the inversion system in implementing nationwide emission adjustments.

Furthermore, the incorporation of a hypothetical geostationary satellite (such as Sentinel 4) in conjunction with ground-based monitoring stations, could enhance the temporal resolution and enable the capture of daily $NO_x$ cycles, while also increasing the sensitivity of the satellite data near the surface.

*Code availability.* The CHIMERE code is available here: www.lmd.polytechnique.fr/chimere/, (Menut et al., 2013; Mailler et al., 2017). The CIF code is available here: http://community-inversion.eu/index.html (Berchet et al., 2021).

*Data availability.* The re-processed TROPOMI-PAL dataset is available on https://data-portal.s5p-pal.com (Eskes et al., 2021). The CAMS-REG inventory (Kuenen et al., 2022) is available upon request from TNO (contact: Hugo Denier van der Gon, hugo.deniervandergon@tno.nl).

The CITEPA monthly budgets for France are available on https://www.citepa.org/fr/barometre/. The INS inventory is available on http://emissions-air.developpement-durable.gouv.fr/indexMap.html.

## Appendix A: Metropolitan masks

The list of all 15 urban areas and their masks is displayed on the following figure. Each mask is made out of $10\times10~\mathrm{km}^2$ pixels.

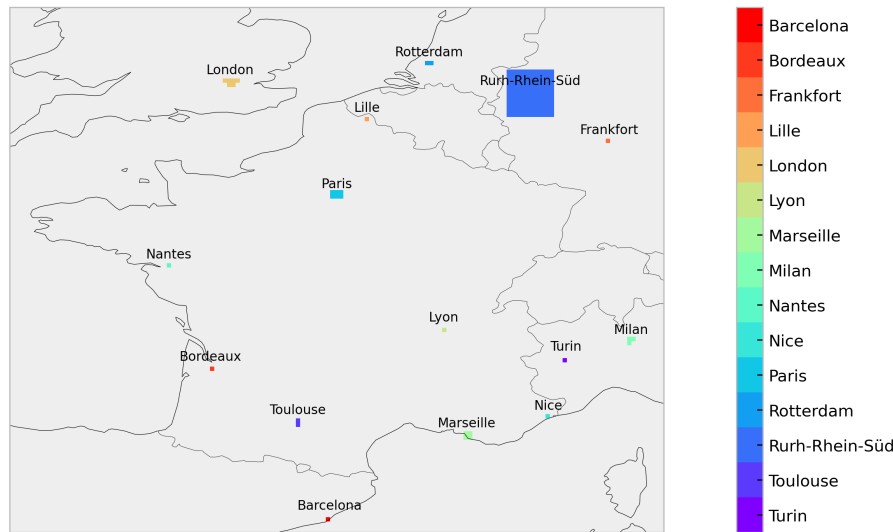

**Figure A1.** *Urban areas studied.*

| Urban area | Area [number of 10×10 km$^2$ pixels] | Population density [inhab.km$^{-2}$] EuroStat |
|---|---|---|
| Barcelona | 1 | 723 |
| Bordeaux | 1 | 164 |
| Frankfurt | 1 | 633 |
| Lille | 1 | 455 |
| London | 6 | 1435 |
| Lyon | 1 | 580 |
| Marseille | 4 | 289 |
| Milan | 3 | 1591 |
| Nantes | 1 | 209 |
| Nice | 1 | 253 |
| Paris | 6 | 1025 |
| Rotterdam | 2 | 1092 |
| Ruhr-Rhein-Süd | 121 | - |
| Toulouse | 2 | 221 |
| Turin | 1 | 329 |

**Table A1.** *Cities studied in the paper and their areas in pixels with their population density according to EuroStat (statistics for 2019).*

## Appendix B: Additional tables

| Domain or region | Period | CAMS-REG/INS + MEGAN inventories | Posterior in 2019 | | Posterior in 2020 | | Posterior in 2021 | | Posterior 2020-2019 | Posterior 2021-2020 |
|---|---|---|---|---|---|---|---|---|---|---|
| | | kteqNO$_2$ | kteqNO$_2$ | [%] | kteqNO$_2$ | [%] | kteqNO$_2$ | [%] | [%] | [%] |
| Paris | Annual | 27 | 24 | -11 | 23 | -17 | 23 | -15 | -6 | +2 |
| | Spring (MAM) | 7 | 7 | -7 | 6 | -21 | 6 | -14 | -15 | +9 |
| | March | 3 | 2 | -14 | 2 | -24 | 2 | -17 | -11 | +9 |
| | April | 3 | 2 | -6 | 2 | -30 | 2 | -17 | -25 | +19 |
| | May | 2 | 2 | -1 | 2 | -7 | 2 | -7 | -7 | +1 |
| | November | 2 | 2 | -21 | 2 | -15 | 2 | -18 | +8 | -4 |

**Table B1.** *Prior and posterior NO$_x$ total emission budgets in* kteqNO$_2$ *and their relative differences in %* $[100 \times (posterior - prior)/prior]$*, in Paris for different periods. Columns "Posterior year$_n$ − year$_{n-1}$" show the relative difference between year$_n$ and year$_{n-1}$ posterior fluxes in %* $[100 \times (F_{year_n} - F_{year_{n-1}})/F_{year_{n-1}}]$*.*

| Urban area | TROPOMI TVCD | | CHIMERE prior TVCD | | CHIMERE posterior TVCD | | Emissions posterior | |
|---|---|---|---|---|---|---|---|---|
| | Mar | Apr | Mar | Apr | Mar | Apr | Mar | Apr |
| Barcelona | -42 | -72 | -13 | -12 | -20 | -29 | -4 | -8 |
| Bordeaux | -34 | -27 | -14 | +23 | -15 | +21 | -3 | -1 |
| Frankfurt | -54 | +1 | -18 | +58 | -25 | +39 | -7 | -10 |
| Lille | -15 | -35 | +19 | -10 | +4 | -28 | -9 | -11 |
| London | -7 | -38 | +4 | +5 | -3 | -14 | -5 | -12 |
| Lyon | -27 | -43 | +13 | +18 | +1 | +1 | -15 | -13 |
| Marseille | -20 | -27 | -2 | +46 | -9 | +19 | -2 | -7 |
| Milan | -30 | -21 | +28 | +2 | -8 | -13 | -13 | -2 |
| Nantes | +2 | -45 | +16 | -2 | +14 | -7 | -3 | -3 |
| Nice | -28 | -37 | 0 | +7 | -3 | +2 | 0 | -1 |
| Paris | -41 | -54 | -4 | +42 | -12 | -9 | -11 | -26 |
| Rotterdam | -35 | -17 | -15 | +39 | -19 | +12 | -5 | -6 |
| Ruhr-Rhein-Süd | -25 | -9 | -21 | +42 | -24 | +8 | -16 | -26 |
| Toulouse | -41 | -31 | -23 | -23 | -22 | -22 | 0 | +1 |
| Turin | -33 | -41 | +17 | -5 | +1 | -18 | -1 | -3 |

**Table B2.** *Changes in NO$_2$ TROPOMI-PAL and CHIMERE tropospheric columns (in %) and changes in NO$_x$ total posterior emissions (in %), from March/April 2019 to March/April 2020 for the selection of urban areas displayed in Figure A1.*

| Urban area | Standard posterior emissions | | Filtered posterior emissions | | Standard posterior emissions **R**-perfect-model | | Filtered posterior emissions **R**-perfect-model | |
|---|---|---|---|---|---|---|---|---|
| | Mar | Apr | Mar | Apr | Mar | Apr | Mar | Apr |
| Barcelona | -4 | -8 | -7 | -16 | -6 | -13 | -10 | -23 |
| Bordeaux | -3 | -1 | -9 | -2 | -5 | -4 | -13 | -7 |
| Frankfurt | -7 | -10 | -12 | -17 | -11 | -21 | -20 | -36 |
| Lille | -9 | -11 | -16 | -11 | -10 | -15 | -17 | -13 |
| London | -5 | -12 | -3 | -18 | -5 | -16 | -2 | -23 |
| Lyon | -15 | -13 | -24 | -24 | -15 | -25 | -22 | -41 |
| Marseille | -2 | -7 | -5 | -9 | -2 | -10 | -5 | -13 |
| Milan | -13 | -2 | -22 | -11 | -14 | -4 | -22 | -14 |
| Nantes | -3 | -3 | -2 | -7 | -5 | -5 | -6 | -11 |
| Nice | 0 | -1 | -4 | -3 | -1 | -4 | -5 | -7 |
| Paris | -11 | -26 | -15 | -28 | -13 | -31 | -15 | -32 |
| Rotterdam | -5 | -6 | -6 | -10 | -5 | -9 | -3 | -12 |
| Ruhr-Rhein-Süd | -16 | -26 | -16 | -26 | -16 | -31 | -15 | -31 |
| Toulouse | 0 | +1 | -1 | +1 | -1 | +1 | -4 | -1 |
| Turin | -1 | -3 | -10 | -4 | +1 | -6 | -7 | -8 |

**Table B3.** *Changes in $NO_x$ CHIMERE total posterior emissions for the standard (in %) and with a simplified observation error set-up (in %, **R** matrix without model error, see details in Section 3.2.3 from March/April 2019 to March/April 2020 considering different TROPOMI coverage for the selection of urban areas displayed in Figure A1.*

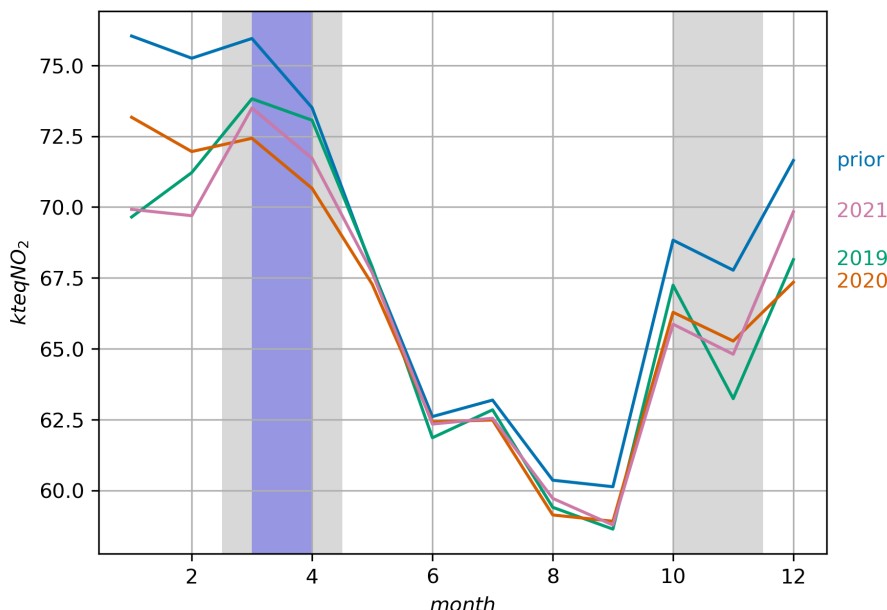

**Figure C1.** *Monthly $NO_x$ anthropogenic emissions in France as estimated by the CAMS-REG/INS inventory in 2016 (in blue) and by the inversions for years 2019 (in green), 2020 (in orange) and 2021 (in pink), in* $\mathrm{kteqNO_2.month^{-1}}$. *Grey shaded areas show the French lockdown periods for the year 2020 and the purple shaded area shows the French lockdown period for the year 2021.*

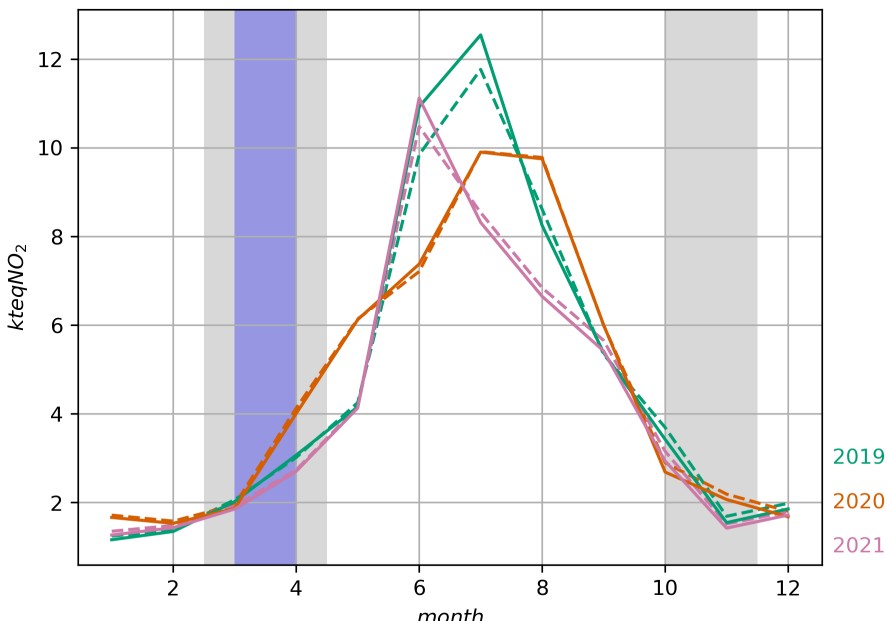

**Figure C2.** *Monthly NO$_x$ biogenic emissions in France as estimated by the MEGAN inventory (dotted lines) and by the inversions for years 2019 (in green), 2020 (in orange) and 2021 (in pink), in* kteqNO$_2$.month$^{-1}$. *Grey shaded areas show the French lockdown periods for the year 2020 and the purple shaded area shows the French lockdown period for the year 2021.*

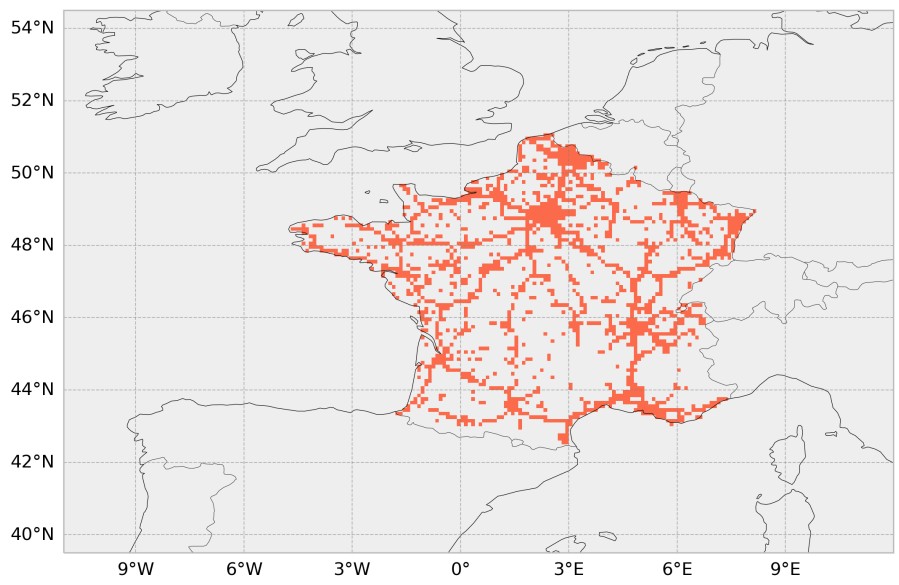

**Figure C3.** *Anthropogenic mask used over France using an urban and road land-use proxy.*

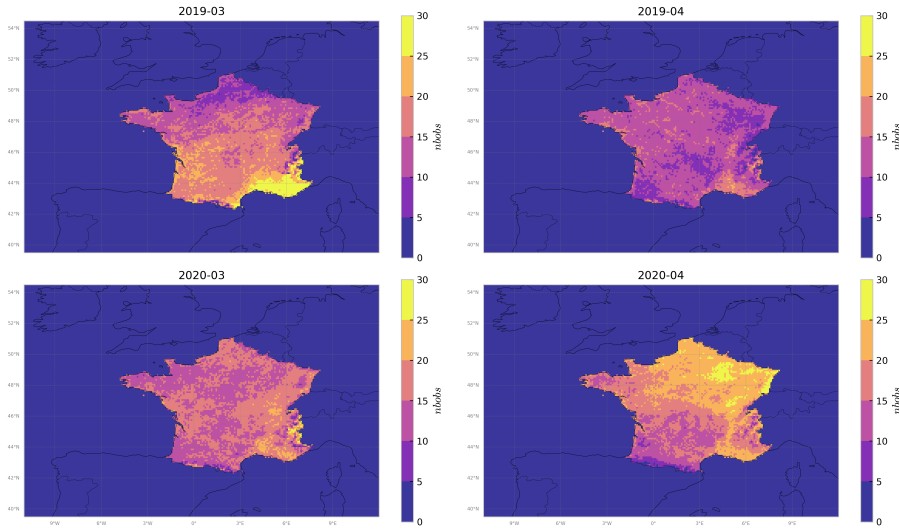

**Figure C4.** *Number of TROPOMI-PAL super-observations on the ARGFR domain in March/April 2019 and March/April 2020.*

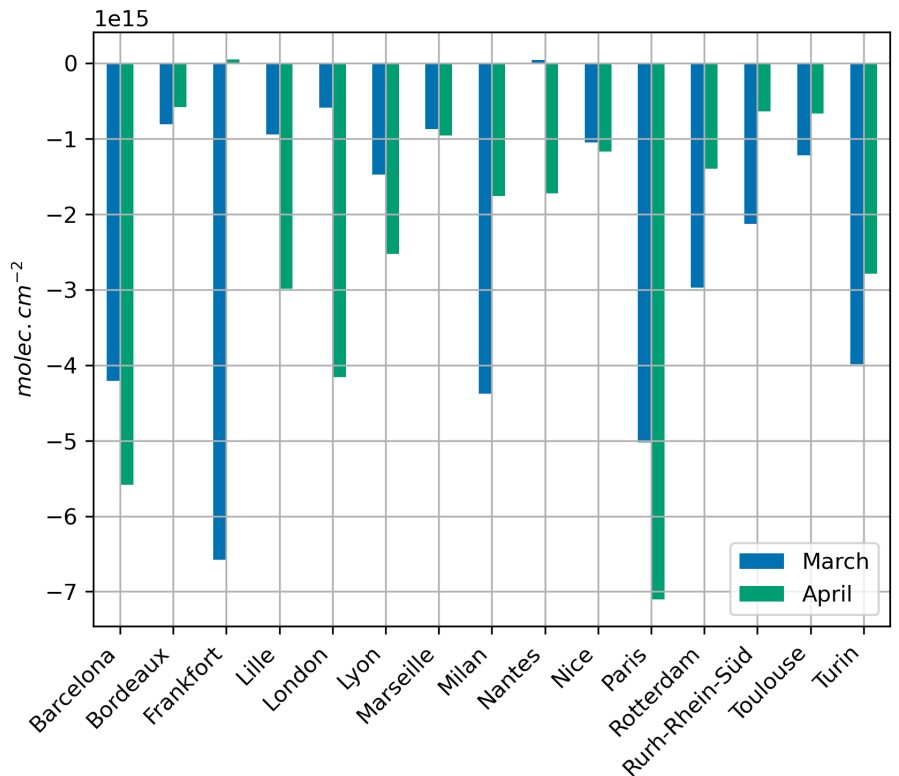

**Figure C5.** *Absolute TROPOMI NO$_2$ TVCD difference between March/April 2019 and March/April 2020 in* molec.cm$^{-2}$ *for the selection of urban areas displayed in Figure A1.*

*Author contributions.* RP, AFC, GB, IP, AB, EP, GD, AC, DS, and GS conceptualized the study. GS produced the prior emissions map. HE produced the satellite data. RP and AFC carried out inversions and data analysis. All co-authors contributed to the design of the study and to writing the manuscript.

*Competing interests.* The authors declare that they have no conflict of interest.

*Acknowledgements.* This study has received funding from the French ANR project ARGONAUT under grant agreement No ANR-19-CE01-0007 and from the French PRIMEQUAL project LOCKAIR under grant agreement No 2162D0010. This work was also supported by the CNES (Centre National d'Etudes Spatiales), in the frame of the TOSCA ARGOS project. This work was granted access to the HPC resources of TGCC under the allocations A0100102201 and A0110102201 made by GENCI. Finally, we wish to thank J. Bruna (LSCE) and his team for computer support.

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
