# Peer review of "$NO_x$ emissions in France in 2019-2021 as estimated by the high spatial resolution assimilation of TROPOMI $NO_2$ observations"

_EGUsphere, 2024_

## Author Comment (AC1)

**Review 1**

The present study assesses the potential of the TROPOMI-PAL NO2 observation to derive NOx emissions in France from 2019 to 2021, using prior official emission data from the year 2016 and the Community Inversion Framework coupled to the CHIMERE regional transport model. The study compares the intra-annual relative changes obtained with the satellite-based emissions against the ones reported by the national bottom-up inventory constructed by CITEPA. At the national scale, the top-down estimates fail to reproduce the relative changes reported by CITEPA due to the COVID-19 restrictions, the inconsistencies being attributed by the authors to limitations in the TROPOMI-PAL NO2 observation coverage and the ratio between the current level of errors in the observation and the chemistry-transport model. The authors perform sensitivity runs to assess the impact of the aforementioned limitations. At the urban scale, where anthropogenic NOx emissions dominate, and considering only days during which the TROPOMI coverage is good, the relative changes reported by TROPOMI-based estimates are larger and more in line with national drops reported by CITEPA. The paper is well written and structured, which makes it a good contribution to ACP. However, there are some aspects related to the description of the methods and discussion of the results that should be better clarified before the manuscript is accepted for publication.

**We wish to thank the referee for his/her helpful comments. His/her full comments are copied hereafter in normal black font, and our responses are inserted in between in bold font.**

**Prior estimates of the emission maps**

The description of this section is a bit ambiguous and should be clarified. Authors mention that the priors are based on CAMS-REG (year 2016) and the INS inventory (year 2012). However, it is not clear which inventory is being used for which country, pollutant sector and species. From the current text, I assumed INS is being used in France and CAMS-REG in the other countries contained in the CHIMERE working domain. However, later in the presentation and discussion of the results (e.g. section 3.1.2) the authors keep mentioning that 2016 is the reference year of the prior emissions, from which I assume that it is in fact CAMS-REG the inventory used for France. Could you clarify this point in the text?

**Yes, this section was a bit confusing. We have sorted and clarified this description.**

**The national scale budget of NOx (NO and NO2) emissions and those of 15 other species (including NMVOC and CO) in France in this study are based on the CAMS-REG inventory for the year 2016. The INS inventory proxies for 2012 are used for the horizontal spatialization of the emissions within France. Outside France, the CAMS-REG gridded inventory for 2016 is used directly as the prior estimate of both the national scale budget and spatial distribution at the scale of the model grid cells of the anthropogenic emissions of NOx and other species. Inside and outside France, the temporal and vertical profiles of the emissions are derived from the CAMS-REG inventory.**

Also related to this section:

- How is the split of total NOx emissions into NO and NO2 performed?

**Following the GENEMIS recommendations (Kurtenbach et al., 2001; Aumont et al., 2003), the NOx emissions have been speciated as 90 % of NO, 9.2 % of NO2, and 0.8 % of nitrous acid (HONO) emissions. This information has been added to Section 2.1.**

- Can you provide a reference that describes the spatialization and proxies used for the INS emissions?

**To our knowledge, there is no specific reference describing the proxies used for the INS emissions. Nevertheless, we now refer to the INS website (http://emissions-air.developpement-durable.gouv.fr/)**

where numerous maps are shown and where we can find indications regarding the compilation of these maps.

Prior uncertainty in the NOx emissions

Could you describe in more detail how prior NOx emission uncertainties are defined and which is the uncertainty range assumed? Are the uncertainties sector-dependent? Do they also include uncertainties in the spatial and temporal distribution, or only in the annual totals? Could it be that part of the mismatch between top-down and bottom-up 2019/2020 emission relative differences are related to an issue with the definition of the prior emission uncertainty?

**The presentation of the configuration of the B matrix in section 2.5 has been improved to make it easier to understand. As already described in the text, the inversions provide corrections to total NOx anthropogenic and biogenic emissions (keeping the NO2 to NO emission ratio) with a temporal resolution of 1 day, for the 166 (longitude)×122 (latitude) individual horizontal grid cells of the model with a 10 to 50 km resolution, and over three vertical bands (aggregations of model vertical levels to reduce the dimension of the inversion problem on the vertical axis) for the emissions and are not sector-dependent beyond the split between anthropogenic and biogenic emissions. Therefore, the parameters of the diagonal of the B matrix discussed in section 2.5 correspond to these 1-day and 10 to 50 km scales, which is now clarified at the beginning of the description of B. Please note that we have also slightly improved the description of the log-normal distribution (Equation 2).**

**The discrepancy between the anthropogenic emission changes from 2019 to 2020 between the inversion results and independent estimates from inventories are partly connected to the definition of the prior emissions uncertainty (of the B matrix), since we ignore potential spatial and temporal correlations in the prior uncertainties in the anthropogenic emissions at the 1-day and 10 to 50 km resolution.**

**While such correlations are difficult to justify or to diagnose in general, significant spatial and temporal correlations likely arose during lock-down periods due to the national scale measures taken during the COVID-19 crisis. The general decrease of the anthropogenic emissions and of emission factors between 2016 and 2019-2021 could also push for the use of spatial and temporal correlations in B. However, a proper definition of such correlations is highly challenging and was out of the scope of this study. We restrained the inversion configuration to a more conservative option where the direct information from the satellite is not extrapolated in space and time via correlations in the B matrix, and thus where the departure from the prior emission estimates in the inversion are highly sensitive to the observation coverage and emission footprints.**

**Regarding the amplitude of the prior uncertainties: our sensitivity tests highlighted that the configuration of the error covariance matrix R, which characterizes the errors associated with the observations and the chemistry transport model, played a more critical role than that of B in the moderate fit to the observations. The prior uncertainty in the anthropogenic emissions at 1-day and 10 to 50 km resolution was typically set to ~ 74 %-135 % which is relatively high. The absence of spatial and temporal correlations in B also provided much freedom to the inversion to fit the individual satellite observations. However, the very large amplitude of the observation errors which was accounted in R counterbalanced these flexibilities.**

**These points are now better discussed in Section 3.2.4 and in the conclusion.**

Role of natural NO emissions

In the present study, soil NOx emissions are estimated using the MEGAN model, which tend to significantly underestimate this natural fluxes according to numerous studies, especially in agricultural areas (e.g., Oikawa et al., 2015; Almaraz et al., 2018; Sha et al., 2021; Zhu et al., 2023). Could it be that the posterior results are

not capable of capturing the 2016/2019 or 2019/2020 NOx national emission drops due to the fact that the inversion system is increasing prior soil NOx emissions? Is it possible to split the prior and posterior emission estimates between natural and anthropogenic to see the role of the inversion on each source type?

**Thank you for the references about MEGAN. We have included them in the paper when describing the biogenic emissions in Section 2.1.**

**The inversion system controls separately the biogenic and anthropogenic emissions, so that the split between the two types of emissions is directly available from the system. However, the effective distinction of signal from the biogenic vs. anthropogenic emissions in the satellite observation is challenging. Since both types of emissions are controlled at the daily and 10 to 50 km resolution, the inversion system should strongly rely on the configuration of the prior error covariance matrix B with respect to each of the two type of emissions to distinguish between the respective corrections which should be applied to the prior estimates of each of them, and in particular it should strongly rely on the structure of the spatial correlations of the prior uncertainties for this.**

**However, we use a conservative configuration of the B matrix (avoiding an irrelevant characterization of the spatial patterns of errors in the gridded emission estimates) without spatial correlations for the anthropogenic emissions, or with homogeneous and isotropic spatial correlations decreasing with distance with a low (30 km) spatial correlation length scale for the biogenic emissions. As a result, the spatial structure of the prior uncertainties in our inversions is very similar between the anthropogenic and biogenic emission, and in practice, the inversions has a low ability to distinguish between the biogenic and anthropogenic emissions wherever both can be significant, as assumed by the reviewer, especially since the agricultural biogenic emissions have been overlooked in the inversion set-up (as now clarified in the manuscript).**

**This is actually why we merely ignored the partition between the biogenic and anthropogenic emissions when analyzing the results from the inversion in the manuscript, and thus why we focused these analysis on the total NOx emission estimates. We had more confidence in the spatial distribution of the emissions, and in particular in the split between urban, industrial and road areas and the rest of the territory to discriminate between the biogenic and anthropogenic emissions. Indeed, the study revealed that the satellite observation essentially led to local emission corrections in the inversions (i.e. the area of corrections applied to the emission field to fit the observations merely coincided with the field of view of these observations in the inversion process), mainly due to the short lifetime of the NOx. This should limit (even though it does not completely prevent) the risk that the system erroneously apply corrections to the anthropogenic emissions somewhere that should actually be applied to biogenic emissions away.**

**Nevertheless, following this comment, we now provide in the manuscript some illustration of the posterior estimate of the biogenic emissions given by the inversions, and of its weight in the total emission estimate, (see the two new figures C1 and C2 added in the supplementary material). According to the inversions, the weight of the biogenic emissions remain limited and should not impact much the variations of the total NOx emissions at annual and national scales between 2019 and 2021.**

**We have added sentences in the text in Section 3.1.2: "The focus on total emissions is explained by the fact that the distinction between the signal from the biogenic and anthropogenic emissions in the comparisons between the chemistry transport model and the satellite NO2 observations is challenging in the inversion framework. Since both types of emissions are solved at the 1-day and model grid cell resolution, the inversion system relies on the respective amplitude and spatial correlations in the prior uncertainties in the biogenic and anthropogenic emissions (in B) to discriminate between the corrections to be applied to the prior estimates of the two types of emissions. However, our configuration of B yield similar structures of spatial correlations for the two types of emissions. Therefore, our confidence in the split between biogenic and anthropogenic**

**emissions in the posterior emission estimates is low whenever the two types of emissions have comparable levels according to the prior emission estimates".**

**And: "When considering the split between biogenic and anthropogenic emissions from the inversions despite its lack of reliability, the posterior biogenic emission estimates are close to the prior estimates derived from MEGAN (see Figures C1 and C2 in supplement): in particular they do not appear to be stronger in 2020 than in 2019. Therefore, the low decrease (-3 %) of the posterior estimate of the national budget of the total NOx emission in 2020 can hardly be explained by an increase in the biogenic emissions which would compensate for the decrease of the anthropogenic emissions, even though the biogenic emissions due to agriculture have been overlooked in our inversion set-up (see Section 2.1)".**

[Figure]

*Figure C1. Monthly national scale budgets of the NOx anthropogenic emissions in France as estimated by the CAMS-REG/INS inventory in 2016 (in blue) and by the inversions for years 2019 (in green), 2020 (in orange) and 2021 (in pink), in kteqNO2.month$^{-1}$. Grey shaded areas show the French lockdown periods for the year 2020 and the purple shaded area shows the French lockdown period for the year 2021.*

[Figure]

*Figure C2. Monthly national scale budgets of the NOx biogenic emissions in France as estimated by the MEGAN inventory (dotted lines) and by the inversions for years 2019 (in green), 2020 (in orange) and 2021 (in pink), in kteqNO2.month$^{-1}$. Grey shaded areas show the French lockdown periods for the year 2020 and the purple shaded area shows the French lockdown period for the year 2021.*

*Oikawa, P. Y. et al. Unusually high soil nitrogen oxide emissions influence air quality in a high-temperature agricultural region. Nat. Commun. 6, 8753 (2015).*

*Almaraz, M. et al. Agriculture is a major source of NOx pollution in California. Sci. Adv. 4, eaao3477 (2018).*

*Zhu, Q., Place, B., Pfannerstill, E. Y., Tong, S., Zhang, H., Wang, J., Nussbaumer, C. M., Wooldridge, P., Schulze, B. C., Arata, C., Bucholtz, A., Seinfeld, J. H., Goldstein, A. H., and Cohen, R. C.: Direct observations of NOx emissions over the San Joaquin Valley using airborne flux measurements during RECAP-CA 2021 field campaign, Atmos. Chem. Phys., 23, 9669–9683, https://doi.org/10.5194/acp-23-9669-2023, 2023.*

2019/2020 drops in urban areas

The relative 2019/2020 drop in total posterior emissions in urban areas increase (and get closer to national CITEPA estimates) when only considering days during which the TROPOMI coverage is good and assuming no model errors in the covariance matrix. Nevertheless, for some cities the drops of emissions are still quite low, which appears to be inconsistent with the drops in traffic activity reported by CEREMA (https://dataviz.cerema.fr/trafic-routier/) and the drops reported in other studies such as Barré et al. (2021), especially in the cases of Bordeaux, Nice and Toulouse. Can the authors elaborate more on the reasons behind these discrepancies? Could it be, at least partially, related with the city mask considered (using only 1-2 grid cells in some cities may not be representative enough).

**Indeed, the drops of emissions for Bordeaux, Nice and Toulouse are lower than the ones reported in other studies. Since traffic is the main contributor (CITEPA), we have added information about the drops in traffic for the cities of Bordeaux, Nice and Toulouse with a reference to the CEREMA website: "These small changes of emissions over Bordeaux, Nice and Toulouse does not seem consistent with the drops in traffic activity estimated by the Centre d'études et d'expertise sur les risques, l'environnement, la mobilité et l'aménagement CEREMA (CEREMA, 2023)".**

**This discrepancy can be partly due to the city mask considered, but our sensitivity tests have highlighted the limitations of the TROPOMI coverage and the challenges associated to our configuration of the R error covariance matrix. Indeed, even though the results are filtered in Table 3 to take into account corrections only when observations are available over the city, the filtered relative differences show steeper decreases in emissions for Bordeaux (-9 versus -3 %) and Nice (-4 versus 0 %) between March 2019 and March 2020. For Toulouse, it only changes from 0 to -1 % over the same period. The availability of observations is not the only issue, the TROPOMI signal being simply too weak for certain cities: the change in TVCD is very small for Bordeaux, Marseille, Nice and Toulouse between March 2019 and March 2020 (see Figure below).**

**To bring up this argument in the manuscript, we have added this sentence in Section 3.2.1: "The strength of the TROPOMI NO2 signal differs over these 8 French cities. Indeed, the absolute changes in NO2 TVCD in March/April 2020 compared to March/April 2019 are higher for Paris, Lille, Lyon and Nantes (see Figure C5 in Supplementary materials) than for Bordeaux, Marseille, Nice and Toulouse (i.e., $1.\times10^{15}$ molec.cm$^{-2}$ or less). The population density difference between those cities could partly explain such variability (Table A1)."**

[Figure]

*Figure C5. Absolute TROPOMI NO$_2$ TVCD difference between March/April 2019 and March/April 2020 in molec.cm$^{-2}$ for the selection of urban areas displayed in Figure SA.*

Linked to this point, in the abstract authors mention that "*these reductions are particularly pronounced for the largest French urban areas (e.g., -26% from April 2019 to April 2020 in the Paris urban area), consistently with the reduction in the intensity of vehicle traffic reported during the lockdown period*". As mentioned before, this is not precise for all cities (e.g., Toulouse and Bordeaux according to CEREMA data).

**We agree. We have changed the sentence in the abstract: "These reductions are particularly pronounced for the largest French urban areas with high emission levels (e.g., -26 % from April 2019 to April 2020 in the Paris urban area), reflecting reductions in the intensity of vehicle traffic reported during the lockdown period".**

**We have also added the sentence: "However, the system does not show large emission decreases for some of the largest cities in France (such as Bordeaux, Nice, Toulouse), even though they were also impacted by the lockdown measures".**

Results of the work to support the development of bottom-up inventories

In the introduction, the authors emphasize the large uncertainty associated with bottom-up inventories, which come from several elements, including emission factors and spatial and temporal proxies, among others. The authors indicate that the use of observations can complement and support the development of such inventories. I completely agree with this statement. However, in this study I fail to see in which aspects the obtained top-down results are helping to improve or complement the prior estimates, considering that they are not capable of capturing the relative drop in emissions occurred between 2016/2019 and 2019/2020. I think that the limitations encountered in this work and uncertainties associated to the estimation of satellite-based emissions should be better reflected in the abstract and conclusions, as they can be used as a guideline for others. Based on the results of this exercises, what elements are the top-down emission estimation community currently missing to effectively being capable of accurately accounting NOx emissions in space and time and assess the effectiveness of emission abatement policies?

**We agree with the reviewer and we have updated the abstract and conclusion to better reflect this.**

**It is definitely important to emphasize that, due to the challenges of accurately recording anomalies such as COVID-19 at the national level, we have not yet reached the stage of improving the inventory,**

**especially in a country such as France. This is illustrated by the various development paths outlined at the end of the article, which are crucial steps towards achieving this goal.**

**We also emphasize that our results on cities, when we select days with local observations, allow us to obtain results that appear more relevant. These results open positive perspectives regarding the ability to correct inventories, at least at the local level. Ultimately, the inversion approach could provide relatively rapid estimates of events such as COVID-19, without having to wait for the compilation of inventories 1 to 2 years later.**

**We have dedicated the section 3.2.4 to the limitations in our analysis. Both the abstract and the conclusion now better mention these limitations and several perspectives for this work are provided in the conclusion.**

Other comments

- Abstract: "The inversions lead to a decrease of the average emissions over 2019-2021 compared to 2016 of -3% at national scale". I recommend putting "compared to the 2016 prior emission estimates of…"

  **This has been done.**

- Table 3 and Table B3: error in columns names (some of the should be "Apr" instead of "Mar").

  **This has been corrected.**

- Table B1: If no data available for CITEPA, perhaps better to remove the corresponding columns.

  **This has been done.**

- Data availability: please include the CAMS-REG, CITEPA and INS emission datasets.

  **This has been done.**

- Lines 38-40: "For example, at national and annual scales, these uncertainties reach 50-200% depending on the activity sector in the European Monitoring and Evaluation Programme (EMEP) inventory (Kuenen and Dore, 2019)". According to the Informative Inventory Reports provided by Member States, uncertainties in total national NOx emissions are lower than what is reported by the authors. More specifically, and according to Schindlbacher et al. (2021), in most EU countries the uncertainty estimate for total anthropogenic NOx emissions is below 30% (19% in the case of France).

*Schindlbacher, S., Matthews, B., Ullrich, B.: Uncertainties and recalculations of emission inventories submitted under CLRTAP. Technical report CEIP 01/2021. Available at:* [thttps://www.ceip.at/fileadmin/inhalte/ceip/00_pdf_other/2021/uncertainties_and_recalculations_of_emission_inventories_submitted_under_clrtap.pdf](thttps://www.ceip.at/fileadmin/inhalte/ceip/00_pdf_other/2021/uncertainties_and_recalculations_of_emission_inventories_submitted_under_clrtap.pdf)

**We thank the reviewer for this reference. We have included this information in the introduction.**

---

## Author Comment (AC2)

**Review 2**

The paper by Plauchu et al. entitled 'NOx emissions in France in 2019-2021 as estimated by the high spatial resolution assimilation of TROPOMI NO2 observations' explores the use of TROPOMI NO2 data in combination with an inversion framework to estimate the reduction of NOx emissions due to COVID-19 lockdowns in France in 2020, with a focus on several large urban centres. The paper is very comprehensive and well-structured throughout. As the authors note, the initial results show a somewhat limited correction of the posterior by the observations due to several possible reasons. The authors then try two alternative setups that allow for a larger influence of the observations on the posterior emissions. The authors highlight further data needs and developments that could help improve the use of inversions for case studies such as these.

**General comments**

- Interesting and relevant addition to existing literature.
- Comprehensive description of the methodology.
- Overall, the paper is well written.

**We wish to thank the referee for his/her helpful comments. His/her full comments are copied hereafter in normal black font, and our responses are inserted in between in bold font.**

**Specific comments**

- 281-284: Can you comment on the expected influence of using 2016 prior emissions for this inversion study? If prior emissions (for 2016) are systematically higher (~13% based on CITEPA) compared to 2019 or 2020, could this have an effect on the posterior emissions? What if you would have used 2019 prior emissions?

  **Our study reveal the large weight of the prior estimate of the emissions on the inversion results, at least at national scale, driven by the the respective amplitude of the prior uncertainty and observation error covariance matrices (B and R), but also by the limited satellite coverage which is not compensated by an extrapolation of the information from the observations via spatial and temporal correlations in B. Therefore, in a general way, using prior emission estimates derived from inventories for the year 2019 or 2020 rather than for the year 2016 would have led to significantly different inversion results, with lower emission values on average. However, it is difficult to make assumptions on the amplitude of these differences, which would highly vary as a function of the scales and locations, and on their impact on the inter-annual variations of the emissions between 2019 and 2021.**

  **In particular, in experiments when reducing the amplitude of R and when focusing the analysis on cities where the signal from the anthropogenic emission is large, and on periods when the satellite coverage is good, the weight of the prior emission estimate is lower.**

- 369-372: Could the differences between cities also be partly due to different contributions of (heavy) industry to NOx emissions in or around these urban centres?

  **The INS data do not show major differences in terms of sectoral distribution between the main French cities. In all of them, more than 80 % of the emissions are due to 3 non-industrial sectors: road transport, non-industrial combustion and other mobile sources and machinery. The weight of the industrial sector (combustion in the manufacturing industry and combustion in the energy and energy transformation industry) may be higher for some cities, but these cities are not those where the slopes are the smallest. For Bordeaux, Nice, Nantes and Toulouse (cities where the decline between spring 2019 and spring 2020 is small), the share of total emissions by the industrial sector is 4.93 %, 7.15 %, 11.67 % and 11.3 % respectively.**

For Paris, Lyon, Marseille and Lille (cities where the decrease between spring 2019 and spring 2020 is higher) this share is 16.55 %, 15.62 %, 10.49 % and 8.61 % respectively.

We have added a sentence in the manuscript: "These differences between cities cannot be explained by different contributions of industry to NOx emissions in or around these urban areas as INS data do not show major differences in terms of sectoral distribution between the main French cities".

Actually, the differences between cities appear to be driven by the strength of the TROPOMI signal over those cities. As depicted in Figure C5, the difference between TROPOMI NO2 TVCD over those cities between March and April 2019 and March and April 2020 is smaller than cities like Lille, Lyon, Nantes and Paris.

We have added this sentence in section 3.2.1: "The strength of the TROPOMI NO2 signal differs over these 8 French cities. Indeed, the absolute changes in NO2 TVCD in March/April 2020 compared to March/April 2019 are higher for Paris, Lille, Lyon and Nantes (see Figure C5 in Supplementary materials) than for Bordeaux, Marseille, Nice and Toulouse (i.e., $1.\times10^{15}$ molec.cm$^{-2}$ or less)".

- 422: "yielding a more accurate estimate of the COVID-19 delta". The qualification of the alternative method using filtered observations being more accurate has not been made before in the paper. Can you comment on why you conclude that it is more accurate than the initial estimate? Could this lesson be generalized to other studies looking at emission variations at such high spatial and temporal resolution?

This part is not about an alternative method, but about a different way of analyzing the results as long as we do not have stronger insights on the spatial and temporal correlations of the uncertainties in the gridded emission inventories, which we now have better emphasized.

The "accurate" adjective was not the right one. We have replaced the sentence: "… yielding a characterization of the COVID-19 effects which is more consistent with the changes in emissions estimated by the CITEPA".

As detailed in section 3.2.4, some parts of France, including urban areas, suffer from the lack of observations during the COVID-19 period. Locally, for the days when no observation is available, the posterior emission estimates remain close to the prior ones. When considering the budget of the emissions over all days (with and without observations), the amplitude of the corrections to the prior estimate of the emissions driven by the satellite observations is artificially decreased by the lack of corrections during days when there is no satellite observations. Therefore, we filter out these days to better identify the corrections driven by the satellite.

The real alternative method to support a robust direct diagnostic of the monthly to annual emission budgets and changes would be to characterize in the B matrix the actual spatial and temporal correlations of the uncertainties in the gridded inventories with hourly variations used as prior estimate of the emissions by the inversions. Such a characterization would support the extrapolation in space and time of the information obtained locally and for some days from the satellite observation. However, getting suitable insight on such correlations is challenging since the usual correlation models based on assumptions of isotropy, homogeneity in space and time, and of decrease as a function of distance and time likely poorly match the actual derivation and structures of gridded inventories convolved with typical temporal cycles at diurnal to seasonal scales, which explains why a conservative configuration was used for the B matrix in this study (see section 2.5). The challenge is exacerbated when tackling a period such as 2019-2021, with lock-down measures in response to the COVID-19 crisis highly impacting the emissions and thus the structures of uncertainties

in the emission inventories over large spatial scales but limited periods. These sentences have been added in the conclusion.

**We advise stepwise improvements in the configuration of B for future studies: attempting to include some temporally varying spatial and temporal correlations in B despite the current lack of knowledge to support a fully relevant characterization of these correlations, and, in parallel, increased efforts to diagnose the uncertainties in the gridded inventories used for inversions.**

Technical corrections

- 10: Consider replacing "The inversions lead to a decrease…" by "The inversions suggest a decrease…".

  **This has been corrected.**

- 18: "consistently" should be "consistent".

  **We have removed this part of the sentence.**

  20-22: Consider splitting up the first sentence of the introduction, as it is rather long, e.g., "Nitrogen dioxide (NO2) is of great interest due to its important role in many atmospheric processes with strong implications for air quality, health, climate change and ecosystems. NO2 is emitted mainly by road traffic, thermal power plants and industrial activities and produced in the atmosphere by the oxidation of nitric oxide (NO), which is emitted by the same activities".

  **This has been done.**

- 27: "UE" should be "EU".

  **This has been corrected.**

- 32: "reached with since" should be "reached by".

  **This has been done.**

- 167: "Gloabl" should be "Global".

  **This has been corrected.**

- 279: "with emissions higher than 72 kteqNO2 during winter" either add "per month", or "monthly emissions".

  **We have changed the sentence: "… with monthly emissions higher than 72 kteqNO2 during winter and equal to or lower than 66 kteqNO2 during summer".**

- Figure 5: Would it be possible to show the country borders a bit more clearly in these maps?

  **This has been done.**

- 404: "are about 800 kteqNO2". In lines 280-281 an average of 850 kteqNO2 is mentioned.

  **This has been corrected.**

- 430: "randome" should be "random".

  **This has been corrected.**